# INFERENCE-TIME SCALING OF DIFFUSION LANGUAGE MODELS WITH PARTICLE GIBBS SAMPLING

## ABSTRACT

Discrete diffusion models have recently emerged as strong alternatives to autoregressive language models, matching their performance through large-scale training. However, inference-time control remains relatively underexplored. In this work, we study how to steer generation toward desired rewards without retraining the models. Prior methods typically resample or filter *within a single denoising trajectory*, optimizing rewards step-by-step without trajectory-level refinement. We introduce particle Gibbs sampling for diffusion language models (PG-DLM), a novel inference-time algorithm enabling *trajectory-level refinement* while preserving generation perplexity under reward optimization. PG-DLM constructs a Markov chain over full denoising trajectories and applies a conditional sequential Monte Carlo kernel to resample them. We derive theoretical guarantees for convergence, including asymptotic consistency and variance bounds. Within this framework, we further analyze trade-offs across four key axes for inference-time scaling under fixed budgets: iterations, samples, denoising steps, and reward estimation. Our analysis shows scaling iterations achieves the best reward-perplexity trade-off. Empirically, PG-DLM consistently outperforms prior methods using MDLM and LLaDA-8B as base models across a wide range of compute budgets for reward-guided generation tasks including toxicity and sentiment control as well as linguistic acceptability.

## 1 INTRODUCTION

Recent advances in discrete diffusion models have established them as strong alternatives to autoregressive language models for text generation (Austin et al., 2021; Lou et al., 2023; Sahoo et al., 2024; Shi et al., 2024; Zheng et al., 2025; Nie et al., 2025a). By scaling model size and training data, diffusion language models (DLMs) now match or surpass autoregressive large language models (LLMs) on tasks like code generation and mathematical reasoning, as demonstrated by models such as LLaDA-8B (Nie et al., 2025b) and Dream-7B (Ye et al., 2025).

While this progress has focused primarily on *training-time scaling*, which quickly becomes computationally expensive, a complementary and more efficient strategy remains underexplored: steering DLMs at *inference time* toward desired attributes without modifying the underlying model. Examples include generating texts toward high fluency, specific sentiments, or controlled toxicity (Dathathri et al., 2020; Keskar et al., 2019). This is typically formalized as sampling from a reward-weighted posterior: $p^*(\mathbf{x}_0 \mid \mathbf{c}) \propto p_\theta(\mathbf{x}_0 \mid \mathbf{c}) \exp\left(r(\mathbf{c}, \mathbf{x}_0)/\beta\right)$, where $p_\theta(\mathbf{x}_0 \mid \mathbf{c})$ is the pretrained DLM, $r(\mathbf{c}, \mathbf{x}_0)$ is a reward function scoring the output $\mathbf{x}_0$ given prompt $\mathbf{c}$, and $\beta > 0$ controls reward strength (Rafailov et al., 2024; Korbak et al., 2022).

To sample from the reward-weighted posterior at inference time, prior work has explored search-based strategies (Ma et al., 2025) and particle-based methods like best-of-$n$ and sequential Monte Carlo (SMC), including FK Steering (Singhal et al., 2025), which scale by increasing the number of samples. Another line uses predictor-corrector and remasking strategies (Wang et al., 2025; Lezama et al., 2022), scaling via more denoising steps. chg: These methods maintain multiple parallel samples, each following *a single denoising trajectory* $\mathbf{x}_T, \cdots, \mathbf{x}_0$, sampled step-by-step from $t = T$ to $t = 0$, with resampling at intermediate timesteps. They do not perform *trajectory-level refinement*, i.e., iteratively updating entire generations $\mathbf{x}_{0:T}$ across multiple passes. chg: More recent search-based methods (Zhang et al., 2025a; Jain et al., 2025) achieves trajectory-level refinement

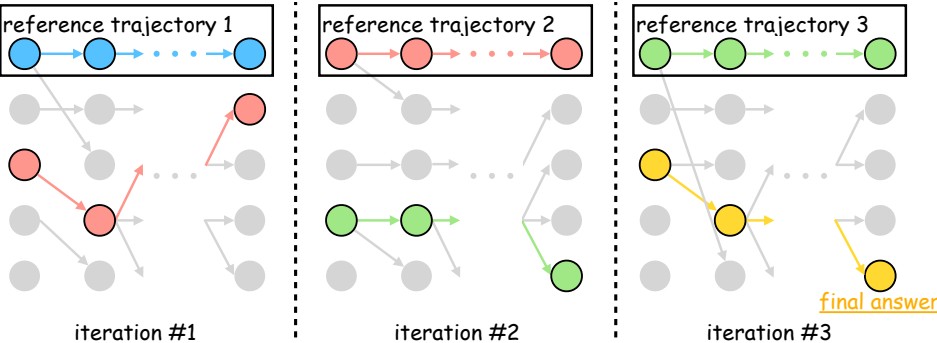

Figure 1: Illustration of PG-DLM. At each iteration, a reference trajectory is fixed (top row), new trajectories are generated and resampled (gray). The highest-reward one becomes the next reference (colored), enabling iterative refinement. The final outputs are selected after multiple iterations.

by revisiting full generations via backtracking in a search tree. In contrast, we introduce the first *particle-based* framework that performs trajectory-level refinement through iterative resampling of complete trajectories within an SMC algorithm, which enables probabilistic inference and adaptive compute allocation.

In this paper, we introduce **particle Gibbs sampling for diffusion language models (PG-DLM)**, a novel inference-time algorithm for reward-guided text generation. Unlike prior chg: particle-based methods that operate step-by-step within a single denoising trajectory, PG-DLM enables *trajectory-level refinement* by iteratively improving full generations. Concretely, PG-DLM runs multiple full generation passes (trajectories) over a sequence of iterations. In each iteration, it generates a batch of trajectories: one trajectory from the previous iteration is fixed as the *reference trajectory*, while the rest are resampled via a conditional sequential Monte Carlo (SMC) kernel, which reweights and resamples at each denoising step based on estimated rewards. The highest-reward trajectory from the current batch then becomes the new reference trajectory for the next iteration.

We further investigate efficient allocation of inference-time compute within PG-DLM. In particular, we analyze trade-offs across four axes: particle Gibbs iterations, samples per iteration, denoising steps, and reward estimation cost. Our analysis shows that scaling samples is most effective in low-compute regimes, but iterations become superior once samples saturate, yielding a better reward-likelihood trade-off by optimizing rewards while preserving generation quality (e.g., perplexity).

Our contributions are threefold: **(1)** we introduce particle Gibbs for diffusion language models (PG-DLM), the first trajectory-level inference-time sampler for discrete DLMs, with formal convergence and variance guarantees (Section 3); **(2)** we develop a unified framework for analyzing inference-time scaling across four axes: iterations, samples, denoising steps, and reward estimation (Section 4); and **(3)** we demonstrate that PG-DLM empirically outperforms baselines like SMC across tasks and budgets (Section 5).

## 2 BACKGROUND

### 2.1 DISCRETE DIFFUSION LANGUAGE MODELS

Discrete diffusion language models (DLMs) (Austin et al., 2021; Lou et al., 2023; Shi et al., 2024; Sahoo et al., 2024) have emerged as a powerful alternative to autoregressive models, matching their performance through large-scale training (Nie et al., 2025b; Ye et al., 2025). Unlike continuous diffusion models (Sohl-Dickstein et al., 2015; Ho et al., 2020; Song & Ermon, 2019), DLMs operate on discrete token spaces, reversing a masking corruption process to iteratively denoise sequences.

Let $\mathbf{x}_0 = (x_1, \ldots, x_L)$ denote a clean sequence of $L$ tokens, where each token $x_i \in \mathcal{X}$ is a one-hot vector; $\mathbf{x}_t$ the corrupted state at time $t \in [0, T]$; and $\mathbf{m}$ the [MASK] token. The forward process $q$ gradually corrupts $\mathbf{x}_0$ by replacing tokens with $\mathbf{m}$:

$$q(\mathbf{x}_t \mid \mathbf{x}_0) = \mathrm{Cat}(\mathbf{x}_t; \alpha_t \mathbf{x}_0 + (1 - \alpha_t)\mathbf{m}), \tag{1}$$

where $\mathrm{Cat}(\cdot)$ denotes the categorical distribution over the vocabulary, and the noise schedule $\alpha_t$ decreases monotonically from $\alpha_0 = 1$ to $\alpha_T = 0$. This enables a closed-form posterior:

$$q(\mathbf{x}_{t-1} \mid \mathbf{x}_t, \mathbf{x}_0) = \begin{cases} \mathrm{Cat}(\mathbf{x}_{t-1}; \mathbf{x}_t), & \mathbf{x}_t \neq \mathbf{m} \\ \mathrm{Cat}\left(\mathbf{x}_{t-1}; \frac{\alpha_{t-1}-\alpha_t}{1-\alpha_t}\mathbf{x}_0 + \frac{1-\alpha_{t-1}}{1-\alpha_t}\mathbf{m}\right), & \mathbf{x}_t = \mathbf{m} \end{cases} \tag{2}$$

To approximate this posterior, DLMs train a denoising model $\mathbf{x}_\theta(\mathbf{x}_t) \in \Delta^{|\mathcal{X}|}$ to predict $\mathbf{x}_0$ from $\mathbf{x}_t$. The resulting backward transition is $p_\theta(\mathbf{x}_{t-1} \mid \mathbf{x}_t) = q(\mathbf{x}_{t-1} \mid \mathbf{x}_t, \mathbf{x}_\theta(\mathbf{x}_t))$. The model is trained by minimizing the negative evidence lower bound (NELBO) to maximize data likelihood:

$$-\log p_\theta(\mathbf{x}_0) \leq \mathcal{L}_{\mathrm{NELBO}} = \mathbb{E}_{q(\mathbf{x}_t \mid \mathbf{x}_0)}\left[\frac{\alpha_{t-1}-\alpha_t}{1-\alpha_t}\log\left(\mathbf{x}_\theta(\mathbf{x}_t)^\top \mathbf{x}_0\right)\right]. \tag{3}$$

## 2.2 REWARD-WEIGHTED GENERATION OF DIFFUSION LANGUAGE MODELS

In this work, we align diffusion language models $p_\theta(\mathbf{x}_0 \mid \mathbf{c})$ with task-specific rewards $r(\mathbf{c}, \mathbf{x}_0)$, where $\mathbf{c}$ is a prompting prefix and $\mathbf{x}_0$ the generated sequence. Examples include generating high-quality text or sentiment control (Dathathri et al., 2020; Keskar et al., 2019). Following Jaques et al. (2017); Ouyang et al. (2022), this can be formalized as a KL-regularized reinforcement learning objective, where we maximize expected reward while remaining close to the base model $p_\theta$:

$$p^*(\mathbf{x}_0 \mid \mathbf{c}) = \arg\max_p \mathbb{E}_{\mathbf{x}_0 \sim p}[r(\mathbf{c}, \mathbf{x}_0)] - \beta\,\mathrm{KL}\left(p(\mathbf{x}_0 \mid \mathbf{c}) \,\|\, p_\theta(\mathbf{x}_0 \mid \mathbf{c})\right), \tag{4}$$

where hyperparameter $\beta > 0$ controls the trade-off between reward maximization and divergence from the base model. This objective has a closed-form solution (Rafailov et al., 2024)

$$p^*(\mathbf{x}_0 \mid \mathbf{c}) \propto p_\theta(\mathbf{x}_0 \mid \mathbf{c}) \cdot \exp\left(r(\mathbf{c}, \mathbf{x}_0)/\beta\right), \tag{5}$$

which reweights the base model distribution toward higher-reward generations. While fine-tuning methods can align base models $p_\theta$ to the target $p^*$ (Clark et al., 2023; Black et al., 2024; Fan et al., 2024; Wallace et al., 2024), we instead pursue *inference-time* approximation via sampling.

## 3 METHOD

In this section, we first derive the reward-weighted generation objective from an RL perspective and present sequential Monte Carlo (SMC) as a baseline sampler. We then introduce particle Gibbs sampling for diffusion language models (PG-DLM), a trajectory-level refinement method that overcomes SMC's limitations, and demonstrate its generality while proving convergence guarantees.

### 3.1 PROBLEM SETUP AND SEQUENTIAL MONTE CARLO FOR DLMS

In the backward process of a DLM $p_\theta(\mathbf{x}_0 \mid \mathbf{c})$, generation begins with a fully masked sequence $\mathbf{x}_T = \mathbf{m}$ and iteratively unmasks tokens via the denoising model $p_\theta(\mathbf{x}_{t-1} \mid \mathbf{c}, \mathbf{x}_t)$, yielding a full *denoising trajectory* $\mathbf{x}_{T:0} = \mathbf{x}_T, \ldots, \mathbf{x}_0$. However, to sample from the reward-weighted target distribution $p^*(\mathbf{x}_0 \mid \mathbf{c})$ as in Equation 5, one must use the corresponding conditional distributions $p^*(\mathbf{x}_{t-1} \mid \mathbf{c}, \mathbf{x}_t)$ at each timestep. Building on prior works in the continuous setting (Uehara et al., 2024a;b), we derive the tractable formulation for these conditionals in the discrete setting:

$$p^*(\mathbf{x}_{t-1} \mid \mathbf{c}, \mathbf{x}_t) \propto p_\theta(\mathbf{x}_{t-1} \mid \mathbf{c}, \mathbf{x}_t) \cdot \exp\left(r(\mathbf{c}, \mathbf{x}_{t-1}) - r(\mathbf{c}, \mathbf{x}_t)\right),$$
$$\text{where } r(\mathbf{c}, \mathbf{x}_t) = \log \mathbb{E}_{p_\theta(\mathbf{x}_0 \mid \mathbf{c}, \mathbf{x}_t)}\left[\exp\left(r(\mathbf{c}, \mathbf{x}_0)/\beta\right)\right]. \tag{6}$$

Here, $r(\mathbf{c}, \mathbf{x}_t)$ defines a *partial reward function* for the noisy intermediate state $\mathbf{x}_t$, representing the expected future reward at timestep $t$ under the pretrained model $p_\theta$. This formulation shows that the conditional $p^*(\mathbf{x}_{t-1} \mid \mathbf{c}, \mathbf{x}_t)$ is a reward-weighted posterior, with weights given by the difference in partial rewards. It mirrors the reward-weighted objective in Equation 5 through timestep-wise decomposition, incorporating the reward difference at each step. chg: While we formally derive the reward-difference structure from an RL perspective, where the difference in rewards across timesteps $r(\mathbf{c}, \mathbf{x}_{t-1}) - r(\mathbf{c}, \mathbf{x}_t)$ is used to guide generation, similar formulations have been used as sampling

---

**Algorithm 1:** Particle Gibbs for Diffusion Language Models

**Input** : iterations $m$, sample count $k$, timesteps $T$, partial reward samples $\phi$, reward model $r(\mathbf{c}, \mathbf{x}_0)$,
    diffusion model $p_\theta(\mathbf{x}_{t-1} \mid \mathbf{c}, \mathbf{x}_t)$, hyperparameter $\beta$
**Output:** sample from $p^*(\mathbf{x}_0 \mid \mathbf{c}) \propto p_\theta(\mathbf{x}_0 \mid \mathbf{c}) \exp\left(r(\mathbf{c}, \mathbf{x}_0)/\beta\right)$

1 **Function** PG-DLM $(p_\theta, r, m, k, T, \phi, \beta)$ :
2     Sample initial reference trajectory $\mathbf{x}'_{T:0} \sim p_\theta(\mathbf{x}_0 \mid \mathbf{c})$ via backward process
3     **for** $iter = 1$ **to** $m$ **do**
4        Initialize $k$ samples $\mathbf{x}_T^{(i)} = \mathbf{m}$ for $i = 1, \ldots, k$, all masked including the reference $\mathbf{x}_T^{(k)}$
5        **for** $t = T$ **to** $1$ **do**
6           Fix reference $\bar{\mathbf{x}}_{t-1}^{(k)} = \mathbf{x}'_{t-1}$
7           Propose $\bar{\mathbf{x}}_{t-1}^{(i)} \sim p_\theta(\mathbf{x}_{t-1} \mid \mathbf{c}, \mathbf{x}_t^{(i)})$ for $i = 1, \ldots, k-1$
8           Estimate partial reward $\hat{r}(\mathbf{c}, \bar{\mathbf{x}}_{t-1}^{(i)}) = \log\left(\frac{1}{\phi} \sum_{j=1}^{\phi} \exp\left(r(\mathbf{c}, \mathbf{x}_0^{(j)})/\beta\right)\right)$ where
    $\mathbf{x}_0^{(j)} \sim p_\theta(\mathbf{x}_0 \mid \mathbf{c}, \bar{\mathbf{x}}_{t-1}^{(i)})$ for all $j = 1, \ldots, \phi$ and $i = 1, \ldots, k$
9           Compute importance weights $\bar{w}_{t-1}^{(i)} = \exp\left(\hat{r}(\mathbf{c}, \bar{\mathbf{x}}_{t-1}^{(i)}) - \hat{r}(\mathbf{c}, \mathbf{x}_t^{(i)})\right)$ for $i = 1, \ldots, k$
10          Normalize $w_{t-1}^{(i)} = \bar{w}_{t-1}^{(i)} / \sum_{j=1}^{k} \bar{w}_{t-1}^{(j)}$ for $i = 1, \ldots, k$
11          Sample with replacement $\mathbf{x}_{t-1}^{(i)} \sim \{\bar{\mathbf{x}}_{t-1}^{(j)}, w_{t-1}^{(j)}\}_{j=1}^{k}$ for $i = 1, \ldots, k-1$
12          Fix $\mathbf{x}_{t-1}^{(k)} = \mathbf{x}'_{t-1}$
13        **end**
14        Compute unnormalized final weights $\bar{w}_0^{(i)} = \exp\left(r(\mathbf{c}, \mathbf{x}_0^{(i)})/\beta\right)$ for $i = 1, \ldots, k$
15        Normalize $w_0^{(i)} = \bar{w}_0^{(i)} / \sum_{j=1}^{k} \bar{w}_0^{(j)}$ for $i = 1, \ldots, k$
16        Update reference $\mathbf{x}'_{T:0} \leftarrow \mathbf{x}_{T:0}^{(i^*)}$ where $i^* = \arg\max_i w_0^{(i)}$
17     **end**
18     **return** *reference sample* $\mathbf{x}'_0$ *or weighted samples* $\{\mathbf{x}_0^{(i)}, w_0^{(i)}\}_{i=1}^{k}$

---

heuristics in prior works (Singhal et al., 2025; Wu et al., 2023) without establishing explicit connections to RL objectives. This grounding not only justifies the partial-reward weighting but also enables extensions to other KL-regularized tasks.

Given the reward-weighted conditional distribution $p^*(\mathbf{x}_{t-1} \mid \mathbf{c}, \mathbf{x}_t)$ as in Equation 6, one intuitive way to generate samples from this target is to first draw samples from the base model $p_\theta(\mathbf{x}_{t-1} \mid \mathbf{c}, \mathbf{x}_t)$ and then resample them based on their reward weights. This backward process, iterated from $t = T$ down to $t = 0$, is known as *sequential Monte Carlo (SMC)* or *particle filtering*, where $p_\theta$ is the *proposal distribution* and $p^*$ the *target distribution* (Naesseth et al., 2019; Doucet et al., 2001).

Concretely, the SMC sampling algorithm proceeds as follows: At timestep $T$, we initialize $k$ samples as masked sequences $\mathbf{x}_T^i = \mathbf{m}$ for $i = 1, \ldots, k$. Then, for each subsequent timestep $t$, the process involves: (1) **proposing** $\bar{\mathbf{x}}_{t-1}$ samples from the proposal distribution $p_\theta(\mathbf{x}_{t-1} \mid \mathbf{c}, \mathbf{x}_t)$ for each $\mathbf{x}_t$; (2) **reweighting** by computing the importance weights $w_{t-1} = \exp(r(\mathbf{c}, \bar{\mathbf{x}}_{t-1}) - r(\mathbf{c}, \mathbf{x}_t))$ as in Equation 6; and (3) **resampling** with replacement from $\bar{\mathbf{x}}_{t-1}$ according to the normalized weights $w_{t-1}$ to form $\mathbf{x}_{t-1}$. This method has also been referred to as Feynman-Kac Steering (Singhal et al., 2025) in the context of reward-weighted generation for diffusion models.

## 3.2 A PARTICLE GIBBS SAMPLER

While SMC provides a simple way to scale inference-time compute by increasing the number of samples, it has several limitations that hinder effective reward alignment in DLMs. chg: Samples evolve as parallel trajectories interacting only via reweighting and resampling, limiting inter-sample correlations between them. Moreover, it performs a "one-shot" approximation in a single backward pass from $t = T$ to $t = 0$ without iterative *trajectory-level refinement*. Finally, SMC is prone to weight degeneracy and high variance in importance weights under skewed reward landscapes (Naesseth et al., 2019).

To address these limitations, we propose an iterative trajectory-level sampling framework called **particle Gibbs for diffusion language models (PG-DLM)**. Intuitively, as shown in Figure 1, PG-DLM refines high-reward trajectories across multiple sequential denoising processes: we begin by

generating a batch of candidate trajectories $\mathbf{x}_{0:T}$, select the highest-reward one as a "reference trajectory", and then resample new trajectories guided by this reference, exploring variations around it. This process is repeated iteratively, correlating samples across multiple denoising passes and leveraging the full capacity of $p_\theta$. As shown later, this yields better reward optimization while maintaining generation likelihoods.

Formally, PG-DLM is a particle Gibbs sampler (Andrieu et al., 2010), a Markov Chain Monte Carlo (MCMC) algorithm that iteratively refines complete trajectories $\mathbf{x}_{0:T}$. It uses a *conditional sequential Monte Carlo (SMC)* transition kernel to update the trajectories. Here, we refer to "iteration" as a *trajectory-level update* ($m$ iterations) and "timestep" as the denoising steps within a single trajectory ($t = T, \ldots, 0$). As detailed in Algorithm 1, PG-DLM begins by generating one sample from the base model as an initial reference trajectory (line 2), then performs $m$ iterations of conditional SMC updates (lines 3–18). In each iteration, the conditional SMC update proceeds backward through each timestep $t$ by: (1) **fixing** the reference trajectory deterministically as the $k$-th sample (line 7); (2) **proposing** $k - 1$ new samples from the base model (line 8); (3) **reweighting** all $k$ samples, including the fixed $k$-th one (lines 9-11); and (4) **resampling** the first $k-1$ candidates with replacement, proportional to their normalized weights, while keeping the $k$-th sample fixed (lines 12-13). After each iteration, the new reference trajectory is updated to the highest-weighted one from the current batch (lines 15-17). This iterative process allows the final trajectory to closely approximate the target distribution $p^*(\mathbf{x}_0 \mid \mathbf{c})$.

### 3.3 COMPATIBILITY WITH VARIOUS DIFFUSION PROCESSES

The PG-DLM framework is broadly compatible with arbitrary backward transitions $p(\mathbf{x}_{t-1} \mid \mathbf{c}, \mathbf{x}_t)$ in discrete diffusion models. Examples include the standard unmasking in MDLM (Sahoo et al., 2024) (Equation 2), greedy low-entropy unmasking in LLaDA (Nie et al., 2025b), and correction/re-masking mechanisms (Wang et al., 2025; Lezama et al., 2022).

### 3.4 THEORETICAL ANALYSIS

For PG-DLM, convergence depends on accurately computing the importance weights. As shown in Algorithm 1, we approximate the partial reward using $\phi$ Monte Carlo samples $\mathbf{x}_0 \sim p_\theta(\mathbf{x}_0 \mid \mathbf{c}, \mathbf{x}_t)$.

**Lemma 1** *chg: Let $p^*(\mathbf{x}_0 \mid \mathbf{c}) \propto p_\theta(\mathbf{x}_0 \mid \mathbf{c}) \cdot \exp\left(r(\mathbf{c}, \mathbf{x}_0)/\beta\right)$ be the target distribution, where $p_\theta(\mathbf{x}_0 \mid \mathbf{c})$ is a discrete diffusion model with $T$ denoising steps.[1] By the law of large number, the partial reward estimator $\hat{r}(\mathbf{c}, \mathbf{x}_t) = \log \frac{1}{\phi} \sum_{j=1}^{\phi} \left[\exp\left(r(\mathbf{c}, \mathbf{x}_0^{(j)})/\beta\right)\right]$ (cf. Equation 6) converges to the true value as $\phi \to \infty$, when $\mathbf{x}_0^{(j)} \sim p_\theta(\mathbf{x}_0 \mid \mathbf{c}, \mathbf{x}_t)$ are sampled via $t$ denoising process.*

The reference trajectory in PG-DLM ensures that the conditional SMC updates leave the target distribution *invariant* and *ergodic* for $k \geq 2$ (Andrieu et al., 2010). Under standard assumptions for particle Gibbs, and combined with Lemma 1, chg: this directly yields Theorem 1 on asymptotic consistency (adapted from Andrieu et al. (2010)) and Theorem 2 on variance bounds (adapted from Andrieu et al. (2010); Chatterjee & Diaconis (2018)).

**Theorem 1 (Asymptotic Consistency)** *Given Lemma 1, the empirical distribution produced by PG-DLM converges almost surely to the target $p^*(\mathbf{x}_0 \mid \mathbf{c})$ as $m \to \infty$, $\phi \to \infty$, given $k \geq 2$.*

**Theorem 2 (Variance Bound)** *Given Lemma 1, let the unnormalized target be $\tilde{p}(\mathbf{x}_{0:T} \mid \mathbf{c}) = \gamma(\mathbf{c}, \mathbf{x}_0) \cdot p_\theta(\mathbf{x}_{0:T} \mid \mathbf{c})$, where $\gamma(\mathbf{c}, \mathbf{x}_0) = \exp(r(\mathbf{c}, \mathbf{x}_0)/\beta)$. Its normalizing constant is $Z = \sum_{\mathbf{x}_{0:T}} \tilde{p}(\mathbf{x}_{0:T} \mid \mathbf{c})$. For the estimator $\hat{Z}$ from PG-DLM with $k$ samples and $m$ iterations, the variance*

$$\mathrm{Var}(\hat{Z}) \leq \frac{\mathrm{Var}_{p_\theta(\mathbf{x}_0 \mid \mathbf{c})}\left[\gamma(\mathbf{c}, \mathbf{x}_0)\right]}{mk},$$

*where* $\mathrm{Var}_{p_\theta(\mathbf{x}_0 \mid \mathbf{c})}\left[\gamma(\mathbf{c}, \mathbf{x}_0)\right] = \mathbb{E}_{p_\theta(\mathbf{x}_0 \mid \mathbf{c})}[\gamma(\mathbf{c}, \mathbf{x}_0)^2] - Z^2.$

---

[1]chg: For discrete diffusion models defined via continuous-time Markov chains (CTMC), $p_\theta(\mathbf{x}_0 \mid \mathbf{c})$ has no discretization error as $T \to \infty$.

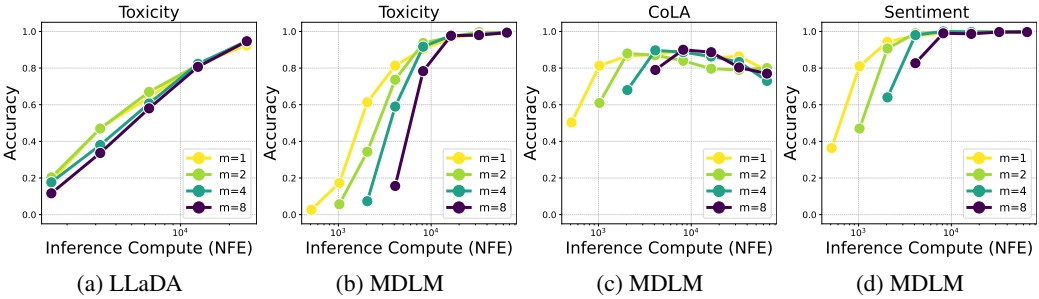

(a) LLaDA  (b) MDLM  (c) MDLM  (d) MDLM

Figure 2: Trade-off between particle Gibbs iterations $m$ and sample counts $k$ across compute budgets (NFEs). The x-axis shows NFEs controlled by varying $k$, and the legend shows $m$. Increasing $k$ (with $m=1$) performs best in low-NFE regimes. However, as samples saturate, additional iterations ($m=2, 4$) become more effective.

| $m$ | $k$ | **Toxicity** |
|---|---|---|
| 1 | 32 | 90.3 |
| 2 | 16 | **93.6** |
| 4 | 8 | 91.7 |
| 1 | 64 | 96.3 |
| 2 | 32 | 97.0 |
| 4 | 16 | **97.6** |

Table 1: Accuracy at high NFE.

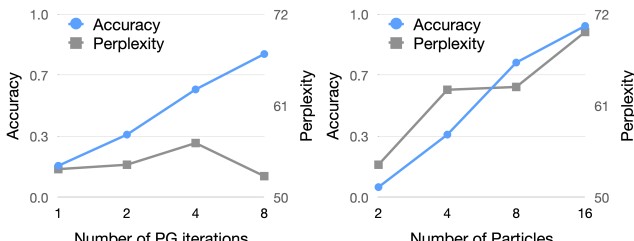

Figure 3: Toxicity accuracy (blue) and perplexity (gray) as compute budgets increase, by varying iterations $m$ (left) and samples $k$ (right)

This variance bound shows that PG-DLM's variance is determined by that of the reweighting function $\gamma(\mathbf{c}, \mathbf{x}_0) = \exp(r(\mathbf{c}, \mathbf{x}_0)/\beta)$ under the proposal $p_\theta(\mathbf{x}_0 \mid \mathbf{c})$. For example, if $r(\mathbf{c}, \mathbf{x}_0)$ is constant, the proposal matches the target and $\mathrm{Var}(\widehat{Z}) = 0$; if $r(\mathbf{c}, \mathbf{x}_0)$ is highly peaked, $\gamma(\mathbf{c}, \mathbf{x}_0)$ has large variance, as the proposal fails to cover high-reward regions effectively, leading to inefficient sampling. chg: Lemma 1 holds for discrete diffusion models such as MDLM and LLaDA. However, in practice, we approximate partial rewards using a small number of $\phi$ samples, each generated with only one denoising step. While this deviates from the asymptotic setting, the convergence and variance bounds still provide valuable insight into how PG-DLM's performance scales with different factors, such as $m, k, T, \phi$, which we study empirically in Section 4.

## 4 INFERENCE-TIME SCALING FOR PG-DLM

In the PG-DLM framework (Algorithm 1), we can scale inference-time compute along four axes: the number of particle Gibbs iterations $m$, samples per iteration $k$, denoising steps $T$, and reward estimation samples $\phi$. This flexibility allows effective allocation under fixed budgets, measured in *number of function evaluations (NFEs)* - the total calls to the denoiser and reward model. Assuming the reward model incurs a similar computational cost to the denoiser (as is typical (Singhal et al., 2025; Ma et al., 2025; Puri et al., 2025)), the total NFE is:

$$\mathrm{NFE} = m \cdot k \cdot T \cdot (1 + \phi). \tag{7}$$

If the reward model is lightweight relative to the base model, we can omit the $\phi$ cost, yielding NFE $= mkT$ (as applied in the LLaDA experiments in Section 5). Given a fixed NFE budget, a key question arises: how to effectively allocate compute across these axes? In this section, we explore this question empirically.

**Particle Gibbs Iterations vs. Sample Count.** We start by examining the trade-off between the number of particle Gibbs iterations $m$ and the number of samples $k$ per iteration. Figure 2 shows that increasing $k$ (with $m = 1$) improves accuracies in low-compute regimes. However, once gains from additional samples saturate, scaling iterations ($m = 2, 4$) proves more effective, especially at moderate-to-high budgets (e.g., NFE $\approx 10^4$). See Table 1 for representative results and full details in Appendix C. Although increasing both $m$ and $k$ can boost performance, Figure 3 shows that higher

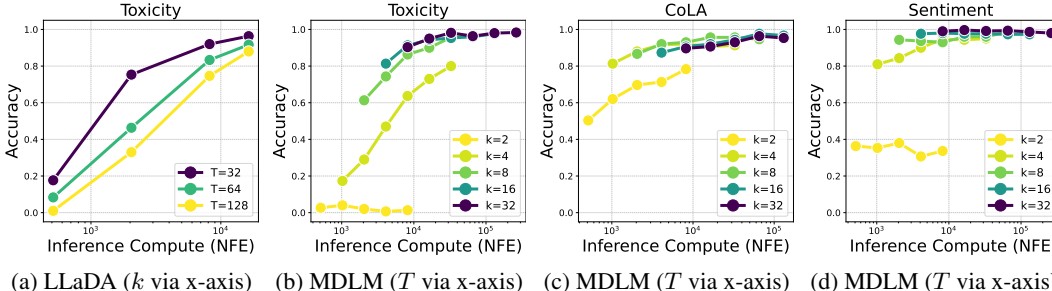

(a) LLaDA ($k$ via x-axis)    (b) MDLM ($T$ via x-axis)    (c) MDLM ($T$ via x-axis)    (d) MDLM ($T$ via x-axis)

Figure 4: Trade-offs between sample counts $k$ and denoising steps $T$ across compute budgets (NFEs). For (a) LLaDA, the x-axis shows NFEs controlled by varying $k$, with $T$ in the legend; for (b-d) MDLM, the x-axis shows NFEs controlled by varying $T$, with $k$ in the legend. Scaling $k$ (and decreasing $T$ accordingly) generally yields better performance under the same NFEs.

$k$ degrades likelihoods (e.g., perplexity) significantly, indicating reward hacking; while higher $m$ keeps likelihoods roughly unchanged. Therefore, scaling $m$ yields a superior reward–perplexity trade-off by enabling iterative trajectory-level refinement without penalizing generation quality.

**Denoising Steps vs. Sample Count.** In masked diffusion models, setting the number of denoising steps $T$ equal to the sequence length $L$ (where at most one token is unmasked per step) is typically sufficient for generation quality, with little benefit from increasing $T$ beyond $L$ (Sahoo et al., 2024). However, this intuition does not hold for PG-DLM. The algorithm performs reward computation and resampling at every timestep, even if no new token is unmasked (Algorithm 1, line 12). Thus, additional steps help concentrate samples closer to the reward-weighted posterior, improving generation quality. This raises the question: Should we prioritize increasing $T$ or the number of samples $k$? To investigate, we first examine compute allocation for LLaDA (Nie et al., 2025b), where $T$ cannot exceed $L$. We fix $L = 128$ and decrease $T$ (from 128 to 64, 32) while increasing $k$ to maintain constant NFEs. We further conduct experiments on standard masked models, generating sequences of length 128 (varying $T$ from 128 to 2048 and $k$ from 2 to 32 accordingly). As shown in Figure 4, increasing $k$ generally provides greater benefits in most cases, chg: though in some cases, e.g., when the performance saturates as in Figure (4c), smaller $k$ can be better. This trend holds across other particle-based methods, including best-of-$n$ and vanilla SMC (Appendix C).

**Partial Rewards Estimation.** To estimate partial rewards $r(\mathbf{c}, \mathbf{x}_t)$ for prompt $\mathbf{c}$ and noisy state $\mathbf{x}_t$, in order to compute importance weights (line 10 in Algorithm 1), we approximate the expectation $\mathbb{E}_{p_\theta(\mathbf{x}_0|\mathbf{c},\mathbf{x}_t)}\left[\exp\left(r(\mathbf{c},\mathbf{x}_0)/\beta\right)\right]$ as in Equation 6 using $\phi$ samples $\mathbf{x}_0 \sim p_\theta(\mathbf{x}_0 \mid \mathbf{c}, \mathbf{x}_t)$ chg: by unrolling $\tau$ diffusion steps per sample. In practice, we set $\tau = 1$ for efficiency following prior works. However, studying the scaling behavior of $\tau$ is an interesting and promising complementary future direction. A common approach is to draw random samples from $p_\theta(\mathbf{x}_0 \mid \mathbf{c}, \mathbf{x}_t)$, yielding unbiased but high-variance estimates (Singhal et al., 2025; Song et al., 2021; Wu et al., 2023; Li et al., 2024). We instead propose *beam sampling* to approximate $p_\theta(\mathbf{x}_0 \mid \mathbf{c}, \mathbf{x}_t)$, with $\phi$ as the beam width, yielding biased but low-variance estimates. For $\phi = 1$, this reduces to greedy decoding. As shown in Figure 5, scaling $\phi$ improves accuracy but raises compute, leading to suboptimal trade-offs. Beam sampling outperforms random methods in most cases, with $\phi = 1$ offering the best trade-off.

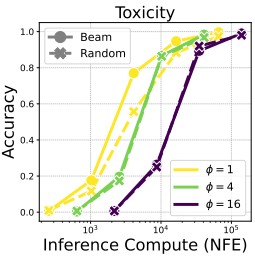

Figure 5: Comparison of Beam and Random sampling for partial reward estimation with varying number of $\mathbf{x}_0$ samples ($\phi$) across NFEs (as controlled by the number of samples $k$). Beam sampling with $\phi = 1$ performs the best.

## 5 EXPERIMENTS

### 5.1 SETUP

We evaluate three reward functions for controllable generation: (1) **Linguistic acceptability**, via a classifier trained on the CoLA dataset, which favors grammatically correct sentences (Morris et al.,

Table 2: Controlled text generation accuracies across reward functions (CoLA, Toxicity, Sentiment) and base models (MDLM, LLaDA), comparing PG-DLM against baselines under varying compute budgets (NFEs). chg: Columns labeled 1–64 correspond to NFEs normalized by the total number of denoising steps $T$, i.e. NFE/$T$.

| Base | Method | CoLA ↑ | | | | Toxicity ↑ | | | | Sentiment ↑ | | | |
|------|--------|------|------|------|------|------|------|------|------|------|------|------|------|
| | | 1 | 4 | 16 | 64 | 1 | 4 | 16 | 64 | 1 | 4 | 16 | 64 |
| MDLM | best-of-$n$ | 27.0 | 71.3 | 96.9 | 95.8 | 0.9 | 1.9 | 11.4 | 33.8 | 10.0 | 36.7 | 79.9 | 99.6 |
| | FK ($\phi=4$) | - | 27.9 | 73.7 | 85.0 | - | 0.8 | 36.6 | 85.9 | - | 10.0 | 86.2 | 98.9 |
| | FK ($\phi=1$) | - | 48.1 | 79.0 | 87.1 | - | **3.8** | 39.8 | 86.1 | - | **37.4** | 91.3 | **99.7** |
| | **PG-DLM** | - | **77.3** | **97.3** | **99.1** | - | 1.4 | **91.1** | **98.1** | - | 23.8 | **96.2** | 99.1 |
| LLaDA | best-of-$n$ | 34.2 | 74.2 | 88.8 | 87.7 | 0.8 | 2.4 | 9.0 | 29.2 | 18.6 | 48.2 | 85.7 | 98.1 |
| | FK | - | 74.1 | 87.9 | 88.2 | - | **9.0** | 43.2 | 80.9 | - | **69.4** | 96.0 | **99.7** |
| | **PG-DLM** | - | **77.8** | **91.1** | **90.6** | - | 8.3 | **48.3** | **89.1** | - | 66.6 | **96.4** | **99.7** |

2020; Warstadt et al., 2019); (2) **Toxicity control**, via a toxicity detector (Logacheva et al., 2022) that identifies harmful content; and (3) **Sentiment control**, via a TweetEval classifier (Barbieri et al., 2020) that steers toward target sentiments (e.g., positive).

We evaluate PG-DLM on two base models: MDLM (Sahoo et al., 2024) and LLaDA-8B-Base (Nie et al., 2025b). We compare against inference-time baselines including best-of-$n$ sampling and FK Steering (FK) (Singhal et al., 2025), whose implementation in prior work is effectively a vanilla SMC algorithm. Following prior work (Singhal et al., 2025; Han et al., 2023), we generate 20 continuations of length 50 for each of 15 controllable generation prompts and report task accuracies on CoLA, Toxicity, and Sentiment. chg: For MDLM, we use 1024 denoising stepes; with best-of-$n$ and FK, we use the vanilla MDLM backward process and resample every 20 steps, as done in (Singhal et al., 2025), while for PG-DLM, we use the ReMDM backward process (Wang et al., 2025) and resample every 5 steps. For LLaDA, we use 50 denoising steps with its native backward decoding and resample every 5 steps for all methods. In all cases, we set $\beta = 0.1$ and the final output is selected as the sample with the highest reward $t = 0$. We report mean performance over 3 random seeds in Table 2 and standard deviations in Table 8. Detailed hyparameters and ablations on these choices are in Appendix D.

## 5.2 RESULTS

Table 2 compares all methods under fixed compute budgets, measured by the number of network function evaluations (NFEs) = $m \cdot k \cdot T \cdot (1 + \phi)$ as in Equation 7, ranging from 1 to 64. Since all methods use the same number of denoising steps $T$ per base model (as detailed in the Setup), we omit it for simplicity in the per-method formulas below.

For MDLM, we account for partial reward estimation, as the reward functions are on the same scale as the base model (millions of parameters). Thus, for best-of-$n$ sampling, NFE equals the number of samples $k$. For FK Steering, NFE is $k \cdot (1+\phi)$, where $\phi$ is the number of $\mathbf{x}_0$ samples used for partial rewards; we show results for $\phi = 1$ and $\phi = 4$ following (Singhal et al., 2025). Unlike Singhal et al. (2025) (which holds $k$ fixed across $\phi$), we adjust $k$ to ensure fair NFE comparisons. For PG-DLM, NFE is $m \cdot k \cdot (1 + \phi)$, accounting for samples $k$, $\phi$ partial reward samples, and iterations $m$. We show results for $m = 1$ and $\phi = 1$ within the current NFE range. Increasing $m$ becomes more effective when $k$ saturates at high NFEs (Section 4).

For LLaDA, we use $\phi = 1$ for partial reward estimation in both PG-DLM and FK Steering, and we omit its cost from the NFE, as the reward functions are lightweight (millions of parameters) relative to the base model (8B). Thus, NFE = $m \cdot k$ for PG-DLM (with $m = 1$ in Table 2) and NFE = $k$ for FK Steering and best-of-$n$ sampling.

Table 2 shows that PG-DLM consistently outperforms baselines on both MDLM and LLaDA across budgets and tasks, highlighting PG-DLM's efficiency in generating high-reward contents.

Table 3: Controlled text generation accuracies (length 512) across reward functions (CoLA, Toxicity, Sentiment) on MDLM, comparing PG-DLM against baselines under varying compute budgets. chg: Columns labeled 1–64 correspond to NFEs normalized by the total number of denoising steps $T$, i.e. NFE/$T$.

| Base | Method | CoLA ↑ | | | | Toxicity ↑ | | | | Sentiment ↑ | | | |
|------|--------|---|---|----|----|---|---|----|----|---|----|----|----|
| | | 1 | 4 | 16 | 64 | 1 | 4 | 16 | 64 | 1 | 4 | 16 | 64 |
| MDLM | best-of-$n$ | 0.0 | 0.3 | 0.0 | 0.3 | 0.3 | 1.0 | 4.3 | 16.7 | 6.0 | 23.0 | 39.7 | 56.3 |
| | FK ($\phi=4$) | – | 0.0 | 0.3 | 5.0 | – | 0.0 | 28.0 | 79.3 | – | 7.3 | 65.3 | 85.0 |
| | FK ($\phi=1$) | – | 0.0 | 2.0 | 6.3 | – | **3.0** | 30.7 | 73.0 | – | **26.0** | 71.0 | 78.7 |
| | **PG-DLM** | – | **34.0** | **62.0** | 58.7 | – | 1.7 | **61.0** | **88.3** | – | 17.3 | **80.0** | **88.7** |

## 5.3 ANALYSIS AND ABLATION

**Longer Sequence Generation.** To assess performance on more challenging inputs, we evaluate controlled generation for sequences of length 512 using 512 denoising steps, while keeping all other settings fixed. As reported in Table 3, the best-of-$n$ baseline shows limited ability to optimize rewards in this regime. In contrast, PG-DLM maintains strong accuracies, with the performance gap widening as the compute budget (NFE) increases.

**Effective Sample Size to Measure Convergence.** We assess the convergence of PG-DLM using the *effective sample size (ESS)*, computed from normalized importance weights $w_i$ for $i = 1, \ldots, k$ at the final timestep of each iteration: ESS $= 1/\sum_{i=1}^{k} w_i^2$. ESS reflects the weight concentration per iteration and ranges from 1 to $k$, with higher values indicating more uniform weights and lower variance. As shown in Table 4, ESS approaches $k$ after a single iteration and continues to increase with more iterations, demonstrating efficient convergence and reduced weight degeneracy.

Table 4: Effective sample size (ESS) for PG-DLM across various number of iterations $m$ and samples per iteration $k$, under a fixed compute budget $m \times k = 64$. chg: ESS is computed per iteration and ranges from 1 to $k$. Results are reported as mean $\pm$ std over multiple runs.

| Setting | Iter 1 | Iter 2 | Iter 3 | Iter 4 | Iter 5 | Iter 6 | Iter 7 | Iter 8 |
|---------|--------|--------|--------|--------|--------|--------|--------|--------|
| $m{=}1, k{=}64$ | $60.2 \pm 5.3$ | – | – | – | – | – | – | – |
| $m{=}2, k{=}32$ | $29.0 \pm 4.1$ | $30.6 \pm 3.1$ | – | – | – | – | – | – |
| $m{=}4, k{=}16$ | $13.3 \pm 3.0$ | $14.9 \pm 2.1$ | $15.2 \pm 1.9$ | $15.5 \pm 1.2$ | – | – | – | – |
| $m{=}8, k{=}8$ | $5.6 \pm 1.9$ | $6.8 \pm 1.8$ | $7.2 \pm 1.5$ | $7.5 \pm 1.3$ | $7.6 \pm 0.9$ | $7.7 \pm 0.8$ | $7.8 \pm 0.5$ | $7.8 \pm 0.6$ |

**The Effect of the Backward Process in Diffusion Models.** We further examine the effect of the backward process by comparing vanilla MDLM dynamics with the recently proposed ReMDM variant (Wang et al., 2025) under different compute budgets. As shown in Figure 6, ReMDM consistently achieves stronger performance, demonstrating our approach's general applicability across different backward processes and its ability to leverage advanced variants for further gains.

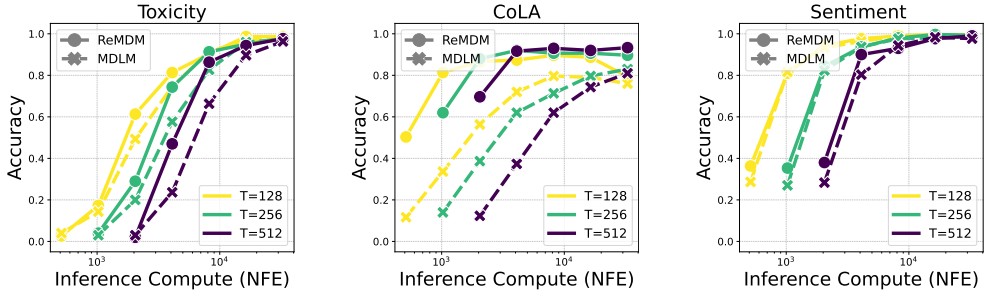

Figure 6: Comparison of ReMDM and vanilla MDLM backward processes under varying compute budgets (NFEs). The x-axis shows NFEs, controlled by varying the number of samples $k$, while the legend shows denoising steps $T \in \{128, 256, 512\}$. ReMDM consistently achieves higher accuracies, demonstrating the effectiveness of improved backward transition dynamics.

### 5.4 CHG: A CASE STUDY ON MATH REASONING TASKS

We evaluate PG-DLM on mathematical reasoning, using LLaDA-8B-Instruct (Nie et al., 2025b) as the base model and testing on GSM8K (Cobbe et al., 2021). We compare against sampling baselines including best-of-$n$, SMC (which we re-implement), and greedy decoding, a common baseline in prior work on math tasks. For all methods, we set the generated length $L = 512$, use $T = 256$ denoising steps, and a block size of 32. For sampling methods, we randomly choose positions to unmask tokens; while for greedy decoding, we deterministically choose the highest-probability position to unmask (Nie et al., 2025a). For SMC and PG-DLM, we resample at the end of each block if the effective sample size (ESS)

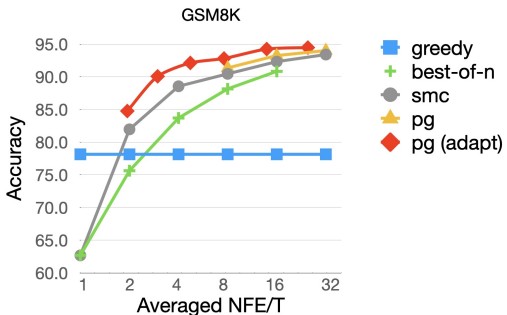

Figure 7: Comparison of all methods under varying compute budgets using LLaDA on GSM8K.

ratio falls below 0.6. We use Qwen2.5-Math-PRM-7B (Zhang et al., 2025b) as the reward model, which has the advantage of computing $r(\mathbf{c}, \mathbf{x}_t)$ directly on partial generations whey they are prefixes, eliminating need to draw samples from $p_\theta(\mathbf{x}_0 \mid \mathbf{c}, \mathbf{x}_t)$.

Additionally, we implement PG-DLM (adapt), a variant that enables adaptive compute allocation through sequential refinement. Starting from a greedy decoding sequence, we perform additional particle Gibbs iterations only when the reward on $\mathbf{x}_0$ is below 0.99. As shown in Figure 7, PG-DLM outperforms SMC at higher NFE, and PG-DLM (adapt) achieves the best accuracy under all compute budget with a significant margin, demonstrating the benefit of trajectory-level refinement.

## 6 RELATED WORK

Inference-time scaling has been extensively studied in autoregressive LLMs, where boosting compute during generation often proves more efficient than training-time scaling (Snell et al., 2024). Techniques like beam search, diverse verifier trees (Beeching et al., 2024), and particle filtering (Puri et al., 2025; Lew et al., 2023) have enhanced mathematical reasoning and constrained generation. While LLMs benefit from these mature tools, analogous strategies for discrete diffusion models remain underdeveloped.

A core approach to scaling diffusion inference is increasing denoising steps: Ma et al. (2025) explore search-based strategies, while Wang et al. (2025) dynamically extend trajectories via re-masking in masked models. chg: For search-based methods, Zhang et al. (2025a); Jain et al. (2025) incorporate mechanisms that can revisit full generation via backtracking in the search tree for trajectory-level refinement, while Guo et al. (2025) performs tree search without explicit refinement of full generations. In contrast, our method perform trajectory-level refinement with resampling-based methods. Particle-based methods scale parallel samples to guide toward high-reward regions (Singhal et al., 2025; Kim et al., 2025), while reinforcement learning optimizes reasoning in diffusion LLMs (Zhao et al., 2025). Predictor-corrector schemes (Lezama et al., 2022; Zhao et al., 2024; Gat et al., 2024) and classifier guidance (Schiff et al., 2025) further improve controllability and quality in discrete settings. In continuous diffusion, particles aid inverse problems (Wu et al., 2023; Dou & Song, 2024; Nazemi et al., 2024) and generation (Kim et al., 2025). Most prior methods apply one-pass sampling within one denoising trajectory, whereas our work performs iterative refinement over multiple trajectories.

## 7 CONCLUSION

We propose a particle Gibbs sampling algorithm for discrete diffusion models that enables efficient inference-time scaling for reward-guided generation. This method iteratively refines full diffusion trajectories, offering theoretical convergence guarantees and strong empirical performance across varying compute budgets, outperforming existing baselines in both quality and scaling behavior.

## ETHICS STATEMENT

All authors have read and adhere to the ICLR Code of Ethics [https://iclr.cc/public/](https://iclr.cc/public/) [CodeOfEthics](https://iclr.cc/public/CodeOfEthics). chg: Controllable generation methods can used to align models with human preferences. Additionally, we recognize that these methods can be used for automated red-teaming, which, if misused, could be used to generate harmful or unsafe content. However, we believe publishing these methods in a transparent and reproducible way enables the research community to better understand behaviors of generative models and develop stronger safeguards. We believe the benefits of this understanding will ultimately outweigh potential risks.

## REPRODUCIBILITY STATEMENT

We present detail experiment setup in Section 5, Appendix C, and Appendix D.

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

# A   SEQUENTIAL MONTE CARLO (SMC)

## A.1   BACKGROUND

**Importance Sampling (IS).** To estimate expectations under a target $f(\mathbf{x})$ (hard to sample from) using a proposal $g(\mathbf{x})$ (easy to sample):

$$\mathbb{E}_f[h(\mathbf{x})] = \mathbb{E}_g\left[h(\mathbf{x})\frac{f(\mathbf{x})}{g(\mathbf{x})}\right] \approx \sum_{i=1}^{N} w_i h(\mathbf{x}^{(i)}), \quad \text{where} \quad w_i = \frac{f(\mathbf{x}^{(i)})}{g(\mathbf{x}^{(i)})}, \{\mathbf{x}^{(i)}\}_{i=1}^{N} \sim g.$$

Resample with replacement via normalized $\{w_i\}$ for approximate samples from $f$.

**Sequential Importance Sampling (SIS).** For sequential targets $f(\mathbf{x}) = \prod_t f(x_t \mid \mathbf{x}_{-1})$ and proposals $g(\mathbf{x}) = \prod_t g(x_t \mid \mathbf{x}_{-1})$, where the full variable is $\mathbf{x} = (x_1, \ldots, x_d)$ and partial prefix $\mathbf{x}_t = (x_1, \ldots, x_t)$ (with $\mathbf{x}_0$ empty), weights factorize recursively:

$$w_t(\mathbf{x}_t) = w_{t-1}(\mathbf{x}_{t-1}) \cdot \frac{f(x_t \mid \mathbf{x}_{t-1})}{g(x_t \mid \mathbf{x}_{t-1})}, \quad w_0 = 1.$$

Propagate $x_t^{(i)} \sim g(\cdot \mid \mathbf{x}_{t-1}^{(i)})$, update $w_t^{(i)}$.

**Sequential Monte Carlo (SMC).** SMC adds resampling to SIS to counter degeneracy. For $N$ particles $\{\mathbf{x}_t^{(i)}, w_t^{(i)}\}_{i=1}^{N}$:

1. Initialize $w_0^{(i)} = 1$.

2. For $t = 1, \ldots, d$:

    (a) Propagate: $x_t^{(i)} \sim g(\cdot \mid \mathbf{x}_{t-1}^{(i)})$.

    (b) Weight: $\tilde{w}_t^{(i)} = w_{t-1}^{(i)} \cdot \frac{f(x_t^{(i)} \mid \mathbf{x}_{t-1}^{(i)})}{g(x_t^{(i)} \mid \mathbf{x}_{t-1}^{(i)})}$.

    (c) Resample $N$ indices $\propto$ normalized $\{\tilde{w}_t^{(i)}\}$; reset to equal weights.

## A.2   SMC FOR DIFFUSION LANGUAGE MODELS

Here we provide pseudocode for vanilla SMC applied to reward-weighted sampling in DLMs, using the conditional $p^*(\mathbf{x}_{t-1} \mid \mathbf{c}, \mathbf{x}_t)$ from Equation 6 as the target and $p_\theta$ as the proposal.

---

**Algorithm 2:** Sequential Monte Carlo for Diffusion Language Models

**Input**  : sample count $k$, timesteps $T$, partial reward samples $\phi$, reward model $r(\mathbf{c}, \mathbf{x}_0)$, diffusion model $p_\theta(\mathbf{x}_{t-1} \mid \mathbf{c}, \mathbf{x}_t)$, hyperparameter $\beta$

**Output:** sample from $p^*(\mathbf{x}_0 \mid \mathbf{c}) \propto p_\theta(\mathbf{x}_0 \mid \mathbf{c}) \exp\left(r(\mathbf{c}, \mathbf{x}_0)/\beta\right)$

1 **Function** SMC-DLM $(p_\theta, r, k, T, \phi, \beta)$:

2     Initialize $k$ samples $\mathbf{x}_T^{(i)} = \mathbf{m}$, all operations on $i$ are over $k$ samples $i = 1, \ldots, k$

3     **for** $t = T$ to $1$ **do**

4        Propose $\bar{\mathbf{x}}_{t-1}^{(i)} \sim p_\theta(\mathbf{x}_{t-1} \mid \mathbf{c}, \mathbf{x}_t^{(i)})$

5        Estimate partial reward $\hat{r}(\mathbf{c}, \bar{\mathbf{x}}_{t-1}^{(i)}) = \log\left(\frac{1}{\phi}\sum_{j=1}^{\phi}\exp\left(r(\mathbf{c}, \mathbf{x}_0^{(j)})/\beta\right)\right)$ where

         $\mathbf{x}_0^{(j)} \sim p_\theta(\mathbf{x}_0 \mid \mathbf{c}, \bar{\mathbf{x}}_{t-1}^{(i)})$ for all $j = 1, \ldots, \phi$

6        Compute importance weights $\bar{w}_{t-1}^{(i)} = \exp\left(\hat{r}(\mathbf{c}, \bar{\mathbf{x}}_{t-1}^{(i)}) - \hat{r}(\mathbf{c}, \mathbf{x}_t^{(i)})\right)$ and normalize

         $w_{t-1}^{(i)} = \bar{w}_{t-1}^{(i)}/\sum_{j=1}^{k}\bar{w}_{t-1}^{(j)}$

7        Sample with replacement $\mathbf{x}_{t-1}^{(i)} \sim \{\bar{\mathbf{x}}_{t-1}^{(j)}, w_{t-1}^{(j)}\}_{j=1}^{k}$

8     **end**

9     Compute final weights $\bar{w}_0^{(i)} = \exp\left(r(\mathbf{c}, \mathbf{x}_0^{(i)})/\beta\right)$ and normalize $w_0^{(i)} = \bar{w}_0^{(i)}/\sum_{j=1}^{k}\bar{w}_0^{(j)}$

10     **return** *argmax sample* $\mathbf{x}_0^{(i^*)}$ *where* $i^* = \arg\max_i w_0^{(i)}$ *or weighted samples* $\{\mathbf{x}_0^{(i)}, w_0^{(i)}\}_{i=1}^{k}$

---

## B  PROOF

### B.1  OPTIMAL DENOISING DISTRIBUTION (EQUATION 6)

Following Uehara et al. (2024b;a), we derive the reward-weighted conditional $p^*(\mathbf{x}_{t-1} \mid \mathbf{c}, \mathbf{x}_t)$ from a per-step KL-regularized RL objective. Define the partial reward $r(\mathbf{c}, \mathbf{x}_t)$ as the expected future reward at timestep $t$:

$$r(\mathbf{c}, \mathbf{x}_t) = \beta \log \mathbb{E}_{\mathbf{x}_0 \sim p_\theta(\mathbf{x}_0 \mid \mathbf{c}, \mathbf{x}_t)} \left[ \exp\left( r(\mathbf{c}, \mathbf{x}_0)/\beta \right) \right]. \tag{8}$$

The optimal conditional maximizes expected partial reward while staying close to the base denoiser:

$$p^*(\mathbf{x}_{t-1} \mid \mathbf{c}, \mathbf{x}_t) = \arg\max_p \mathbb{E}_p \left[ r(\mathbf{c}, \mathbf{x}_{t-1}) \right] - \beta D_{\mathrm{KL}} \left[ p(\mathbf{x}_{t-1} \mid \mathbf{c}, \mathbf{x}_t) \,\|\, p_\theta(\mathbf{x}_{t-1} \mid \mathbf{c}, \mathbf{x}_t) \right]. \tag{9}$$

The solution is tractable:

$$p^*(\mathbf{x}_{t-1} \mid \mathbf{c}, \mathbf{x}_t) \propto p_\theta(\mathbf{x}_{t-1} \mid \mathbf{c}, \mathbf{x}_t) \exp\left( r(\mathbf{c}, \mathbf{x}_{t-1})/\beta \right). \tag{10}$$

Normalizing yields:

$$p^*(\mathbf{x}_{t-1} \mid \mathbf{c}, \mathbf{x}_t) = \frac{p_\theta(\mathbf{x}_{t-1} \mid \mathbf{c}, \mathbf{x}_t) \exp\left( r(\mathbf{c}, \mathbf{x}_{t-1})/\beta \right)}{\sum_{\mathbf{x}'_{t-1}} p_\theta(\mathbf{x}'_{t-1} \mid \mathbf{c}, \mathbf{x}_t) \exp\left( r(\mathbf{c}, \mathbf{x}'_{t-1})/\beta \right)} \tag{11}$$

$$= p_\theta(\mathbf{x}_{t-1} \mid \mathbf{c}, \mathbf{x}_t) \exp\left( \frac{r(\mathbf{c}, \mathbf{x}_{t-1}) - r(\mathbf{c}, \mathbf{x}_t)}{\beta} \right), \tag{12}$$

where the denominator from Equation 11 equals $\exp\left( r(\mathbf{c}, \mathbf{x}_t)/\beta \right)$ by the soft Bellman equation (Theorem 1 of Uehara et al. (2024b)):

$$r(\mathbf{c}, \mathbf{x}_t) = \beta \log \sum_{\mathbf{x}_{t-1}} p_\theta(\mathbf{x}_{t-1} \mid \mathbf{c}, \mathbf{x}_t) \exp\left( r(\mathbf{c}, \mathbf{x}_{t-1})/\beta \right).$$

This yields Equation 6, parallelizing the global RL objective (Equation 4) across timesteps.

### B.2  PROOF OF THE VARIANCE BOUND (THEOREM 2)

Assume the diffusion process incurs no discretization error as $T \to \infty$ and partial reward estimation is accurate as $\phi \to \infty$. Abusing notation, we suppress the fixed conditioning prompt $\mathbf{c}$ (e.g., $p_\theta(\mathbf{x}_0) \equiv p_\theta(\mathbf{x}_0 \mid \mathbf{c})$). Let the proposal be the base model $p_\theta(\mathbf{x}_{0:T}) = p_\theta(\mathbf{x}_T) \prod_{t=1}^{T} p_\theta(\mathbf{x}_{t-1} \mid \mathbf{x}_t)$, and define the reweighting function $\gamma(\mathbf{x}_0) = \exp(r(\mathbf{x}_0)/\beta)$.

The unnormalized target is then

$$\tilde{p}(\mathbf{x}_{0:T}) = \gamma(\mathbf{x}_0) p_\theta(\mathbf{x}_{0:T}),$$

with normalizing constant

$$Z = \sum_{\mathbf{x}_{0:T}} \tilde{p}(\mathbf{x}_{0:T}) = \sum_{\mathbf{x}_{0:T}} \gamma(\mathbf{x}_0) p_\theta(\mathbf{x}_{0:T}) = \mathbb{E}_{p_\theta(\mathbf{x}_0)}[\gamma(\mathbf{x}_0)].$$

The normalized target is $\pi(\mathbf{x}_{0:T}) = \tilde{p}(\mathbf{x}_{0:T})/Z = \gamma(\mathbf{x}_0) p_\theta(\mathbf{x}_{0:T})/Z$, which is essentially $p^*(\mathbf{x}_{0:T})$.

From Andrieu et al. (2010), particle Gibbs variance is bounded by that of the underlying SMC. From Robert et al. (1999); Chatterjee & Diaconis (2018), for the SMC estimator $\widehat{Z}$ with $N$ particles over trajectories $\mathbf{x}_{0:T}$ with proposal $p_\theta(\mathbf{x}_{0:T})$ and target $\pi(\mathbf{x}_{0:T})$,

$$\mathrm{Var}(\widehat{Z}) \leq \frac{Z^2}{N} \left( \exp\left( D_{\mathrm{KL}}(\pi\|p_\theta) \right) - 1 \right),$$

where $\pi$ and $p_\theta$ are defined over $\mathbf{x}_{0:T}$. Now,

$$D_{\mathrm{KL}}(\pi\|p_\theta) = \mathbb{E}_\pi \left[ \log \frac{\pi}{p_\theta} \right] = \mathbb{E}_\pi \left[ \log \frac{\gamma(\mathbf{x}_0)}{Z} \right].$$

By Jensen's inequality,

$$D_{\mathrm{KL}}(\pi\|p_\theta) \leq \log \frac{\mathbb{E}_\pi \left[ \gamma(\mathbf{x}_0) \right]}{Z} = \log \frac{\mathbb{E}_{p_\theta}[\gamma(\mathbf{x}_0)^2]}{Z^2} = \log \frac{\mathbb{E}_{p_\theta(\mathbf{x}_0)}[\gamma(\mathbf{x}_0)^2]}{Z^2}.$$

Thus,

$$\mathrm{Var}(\widehat{Z}) \leq \frac{Z^2}{N} \left( \frac{\mathbb{E}_{p_\theta(\mathbf{x}_0)}[\gamma(\mathbf{x}_0)^2]}{Z^2} - 1 \right) = \frac{\mathbb{E}_{p_\theta(\mathbf{x}_0)}[\gamma(\mathbf{x}_0)^2] - \left( \mathbb{E}_{p_\theta(\mathbf{x}_0)}[\gamma(\mathbf{x}_0)] \right)^2}{N} = \frac{\mathrm{Var}_{p_\theta(\mathbf{x}_0)}(\gamma(\mathbf{x}_0))}{N}.$$

For PG-DLM with $m$ iterations and $k$ samples per iteration ($N = mk$), this yields the stated bound.

# C    ADDITIONAL INFERENCE-TIME SCALING RESULTS FOR SECTION 4

## C.1    HYPER-PARAMETERS

Table 5 summarizes hyper-parameter configurations for the scaling experiments in Section 4. Settings are for PG-DLM, FK Steering (FK), and best-of-$n$ across objectives. Fixed parameters: generated length $L = 128$ for both MDLM and LLaDA (except $L = 50$ for LLaDA in Figure 2); $\beta = 0.1$; and resampling every 5 steps. Rows are grouped by paragraph.

Table 5: Hyper-parameter configurations for scaling experiments.

| Figure | Method | Backward | Partial Reward | Hyper-parameters | | | |
|---|---|---|---|---|---|---|---|
| | | | | $T$ | $m$ | $k$ | $\phi$ |
| **Particle Gibbs Iterations vs. Sample Count** | | | | | | | |
| 2 | PG-DLM | ReMDM | Beam | 128 | 1–8 | 2–256 | 1 |
| 2 | PG-DLM | LLaDA | Beam | 128 | 1–8 | 2–256 | 1 |
| 3 | PG-DLM | ReMDM | Beam | 128 | 1–8 | 2–16 | 1 |
| **Denoising Steps vs. Sample Count** | | | | | | | |
| 4 | PG-DLM | ReMDM | Beam | 128–4096 | 1 | 2–32 | 1 |
| 4 | PG-DLM | LLaDA | Beam | 32–128 | 1 | 2–256 | 1 |
| 8 | FK | MDLM | Random | 128–4096 | – | 2–32 | 1 |
| 9 | FK | MDLM | Random | 128–4096 | – | 2–32 | 4 |
| 10 | best-of-$n$ | MDLM | – | 128–4096 | – | 2–32 | – |
| **Partial Reward Estimation** | | | | | | | |
| 5, 11 | PG | MDLM | Beam, Random | 128 | 1 | 1–256 | 1–16 |

## C.2    ADDITIONAL RESULTS FOR TABLE 1 AND FIGURE 2

Table 6 shows detailed controlled text performance across reward functions (CoLA, Toxicity, Sentiment) under varying compute budgets (NFEs), with different particle Gibbs iterations $m$ and sample counts $k$. Each row fixes NFE while varying $m$ and $k$; best per row bolded. At higher NFEs, increasing $k$ yields diminishing returns, while scaling $m$ is more effective.

Table 6: Controlled text performance across reward functions under varying NFEs, with different $m$ and $k$. Best per row bolded.

| Metric | $m = 1$ | | $m = 2$ | | $m = 4$ | | $m = 8$ | |
|---|---|---|---|---|---|---|---|---|
| | $k$ | Accuracy | $k$ | Accuracy | $k$ | Accuracy | $k$ | Accuracy |
| | 16 | 87.3 | 8 | 87.0 | 4 | **89.7** | 2 | 79.0 |
| | 32 | 89.7 | 16 | 84.0 | 8 | 88.7 | 4 | **90.0** |
| CoLA ↑ | 64 | 85.7 | 32 | 79.7 | 16 | 86.3 | 8 | **88.7** |
| | 128 | **86.3** | 64 | 79.0 | 32 | 83.3 | 16 | 80.3 |
| | 256 | 78.7 | 128 | **80.0** | 64 | 73.0 | 32 | 77.0 |
| | 16 | **81.3** | 8 | 73.7 | 4 | 59.0 | 2 | 15.7 |
| | 32 | 90.3 | 16 | **93.7** | 8 | 91.7 | 4 | 78.3 |
| Toxicity ↑ | 64 | 96.3 | 32 | 97.0 | 16 | **97.7** | 8 | **97.7** |
| | 128 | 98.7 | 64 | **99.7** | 32 | 98.3 | 16 | 98.0 |
| | 256 | 98.7 | 128 | 99.0 | 64 | **99.7** | 32 | 99.3 |
| | 16 | 97.7 | 8 | **99.0** | 4 | 98.0 | 2 | 82.7 |
| | 32 | 99.0 | 16 | 99.7 | 8 | **100.0** | 4 | 99.0 |
| Sentiment ↑ | 64 | 99.7 | 32 | **100.0** | 16 | 99.7 | 8 | 98.7 |
| | 128 | **100.0** | 64 | 99.7 | 32 | 99.7 | 16 | 99.7 |
| | 256 | 99.3 | 128 | 99.7 | 64 | **100.0** | 32 | 99.7 |

### C.3  ADDITIONAL RESULTS FOR FIGURE 4

Figure 4 illustrates trade-offs between sample counts and denoising steps for PG-DLM. Here we show the same trend holds for baselines: sequential Monte Carlo (SMC) (Singhal et al., 2025) and best-of-$n$ (BON), where scaling samples generally outperforms steps under fixed NFEs. We use MDLM as the base model.

1. For SMC with number of $\mathbf{x}_0$ samples $\phi = 1$:

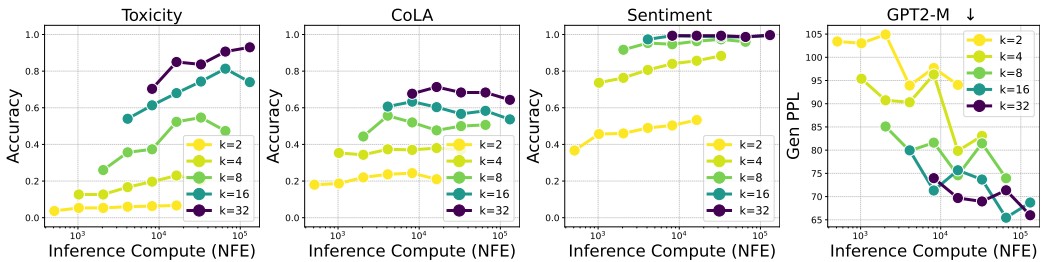

Figure 8: Trade-offs between sample counts $k$ and denoising steps $T$ across compute budgets (NFEs) for **SMC ($\phi = 1$)**. The x-axis shows NFEs controlled by varying $T$, with $k$ in the legend. Scaling $k$ (and decreasing $T$ accordingly) generally yields better performance under the same NFEs.

2. For SMC with number of $\mathbf{x}_0$ samples $\phi = 4$:

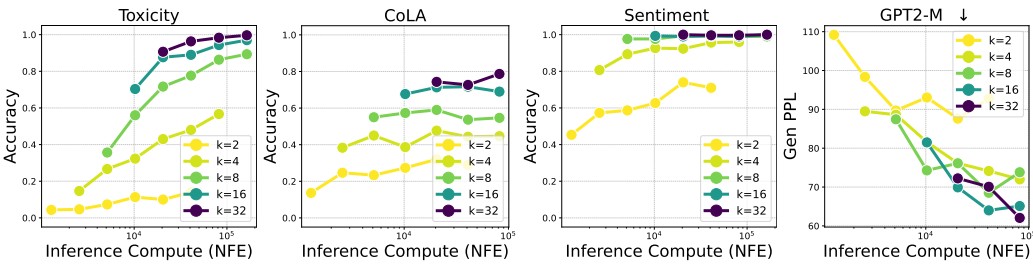

Figure 9: Trade-offs between sample counts $k$ and denoising steps $T$ across compute budgets (NFEs) for **SMC ($\phi = 4$)**. The x-axis shows NFEs controlled by varying $T$, with $k$ in the legend. Scaling $k$ (and decreasing $T$ accordingly) generally yields better performance under the same NFEs.

3. For BON:

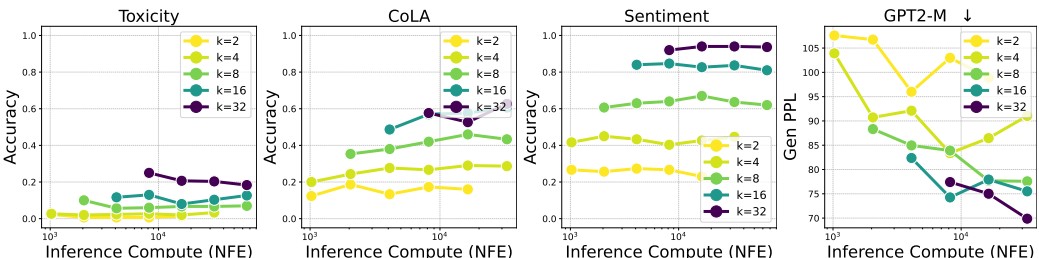

Figure 10: Trade-offs between sample counts $k$ and denoising steps $T$ across compute budgets (NFEs) for **BON**. The x-axis shows NFEs controlled by varying $T$, with $k$ in the legend. Scaling $k$ (and decreasing $T$ accordingly) generally yields better performance under the same NFEs.

### C.4 ADDITIONAL RESULTS FOR FIGURE 5

Figure 11 shows full results for partial reward estimation trade-offs, comparing beam vs. random sampling with varying $\phi$ (samples for $\mathbf{x}_0$ estimation) across NFEs.

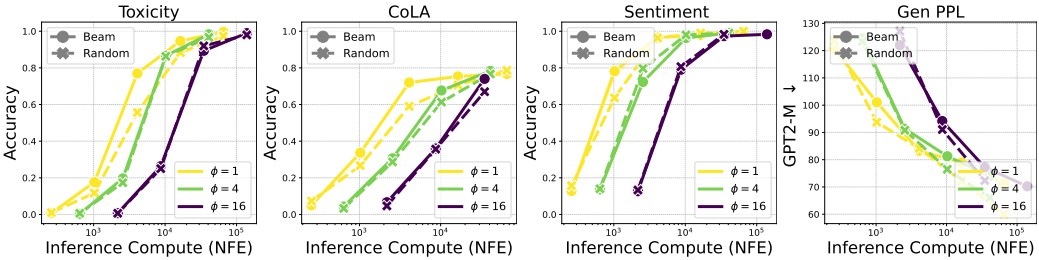

Figure 11: Comparison of Beam and Random sampling for partial reward estimation with varying number of $\mathbf{x}_0$ samples ($\phi$) across NFEs (as controlled by the number of samples $k$). Beam sampling with $\phi = 1$ performs the best.

## D ADDITIONAL EXPERIMENTS RESULTS FOR SECTION 5

### D.1 HYPER-PARAMETERS

Table 7 summarizes hyper-parameter configurations for the experiments in Section 5. Settings are for PG-DLM, FK Steering (FK), and best-of-$n$ across objectives. Hyperparameter include generated text length ($L$), total denoising steps ($T$), particle Gibbs iterations ($m$), sample counts ($k$), the number of $\mathbf{x}_0$ examples for partial reward estimation ($\phi$), and resample frequency ($f$). Rows are grouped by objective.

Table 7: Hyper-parameter configurations for experiments in Section 5

| Table | Method | Base Model | Backward | Partial Reward | $L$ | $T$ | $m$ | $k$ | $\phi$ | f |
|---|---|---|---|---|---|---|---|---|---|---|
| **Conditional Text Generation for MDLM and LLaDA** | | | | | | | | | | |
| 2 | best-of-$n$ | MDLM | MDLM | - | 50 | 1024 | - | $\{1, 4, 16, 64\}$ | - | - |
| 2 | FK ($\phi = 4$) | MDLM | MDLM | Random | 50 | 1024 | - | $\{1, 4, 13\}$ | 4 | 20 |
| 2 | FK ($\phi = 1$) | MDLM | MDLM | Random | 50 | 1024 | - | $\{2, 8, 32\}$ | 1 | 20 |
| 2 | PG-DLM | MDLM | ReMDM | Beam | 50 | 1024 | 1 | $\{2, 8, 32\}$ | 1 | 5 |
| 2 | best-of-$n$ | LLaDA | LLaDA | - | 50 | 50 | - | $\{1, 4, 16, 64\}$ | - | - |
| 2 | FK | LLaDA | LLaDA | Random | 50 | 50 | - | $\{1, 4, 16, 64\}$ | 1 | 5 |
| 2 | PG-DLM | LLaDA | LLaDA | Beam | 50 | 50 | 1 | $\{1, 4, 16, 64\}$ | 1 | 5 |
| **Conditional Text Generation for Longer Sequences** | | | | | | | | | | |
| 3 | best-of-$n$ | MDLM | MDLM | - | 512 | 512 | - | $\{1, 4, 16, 64\}$ | - | - |
| 3 | FK ($\phi = 4$) | MDLM | MDLM | Random | 512 | 512 | - | $\{1, 4, 13\}$ | 4 | 20 |
| 3 | FK ($\phi = 1$) | MDLM | MDLM | Random | 512 | 512 | - | $\{2, 8, 32\}$ | 1 | 20 |
| 3 | PG-DLM | MDLM | ReMDM | Beam | 512 | 512 | 1 | $\{2, 8, 32\}$ | 1 | 5 |

FK Steering (Singhal et al., 2025) reports $\phi = 1$ and $\phi = 4$, but without same-NFE comparisons. We use $\phi = 1$ ($k \in \{2, 8, 32\}$) and $\phi = 4$ ($k \in \{1, 4, 13\}$, adjusted for same-NFE comparison) to match NFEs.

### D.2 REWARD FUNCTIONS AND BASELINES

We evaluate four reward functions for controllable generation:

1. **Linguistic Acceptability**: Favors grammatically correct sentences using a RoBERTa classifier (Morris et al., 2020) trained on CoLA (Warstadt et al., 2019). We measure

    CoLA classification accuracy. Model: `https://huggingface.co/textattack/roberta-base-CoLA`.

2. **Controlled Toxicity**: Guides toward (or away from) toxic outputs using a RoBERTa toxicity classifier (Logacheva et al., 2022) for red-teaming. We measure toxicity classification accuracy. Model: `https://huggingface.co/SkolkovoInstitute/roberta_toxicity_classifier`.

3. **Controlled Sentiment**: Steers toward target sentiments (e.g., positive) using a RoBERTa classifier (Barbieri et al., 2020) on TweetEval. We measure sentiment classification accuracy. Model: `https://huggingface.co/cardiffnlp/twitter-roberta-base-sentiment`.

4. **Perplexity**: Encourages fluency by minimizing perplexity computed by GPT2-Small (Radford et al., 2019). We evaluate using generative perplexity under GPT2-XL. Model: `https://huggingface.co/openai-community/gpt2`.

Baseline implementations for FK Steering and best-of-$n$ are adapted from `https://github.com/zacharyhorvitz/Fk-Diffusion-Steering/tree/main/discrete_diffusion`; we re-ran experiments for consistency.

### D.3 STANDARD DEVIATION OF TABLE 2

Table 8: Standard deviations (±) for controlled text generation metrics in Table 2.

| Base | Method | CoLA ↑ | | | | Toxicity ↑ | | | | Sentiment ↑ | | | |
|------|--------|---|---|----|----|---|---|----|----|---|---|----|----|
| | | 1 | 4 | 16 | 64 | 1 | 4 | 16 | 64 | 1 | 4 | 16 | 64 |
| MDLM | best-of-$n$ | 2.0 | 1.3 | 1.6 | 1.3 | 0.8 | 0.4 | 1.0 | 2.8 | 1.0 | 3.7 | 1.0 | 0.2 |
| | FK ($\phi=4$) | - | 4.5 | 4.1 | 1.2 | - | 0.2 | 1.2 | 1.7 | - | 1.3 | 1.7 | 0.4 |
| | FK ($\phi=1$) | - | 1.6 | 4.3 | 1.9 | - | 1.0 | 3.7 | 1.1 | - | 1.2 | 3.4 | 0.3 |
| | **PG-DLM** | - | 2.0 | 0.9 | 0.5 | - | 0.7 | 1.0 | 1.1 | - | 2.2 | 1.3 | 0.2 |
| LLaDA | BoN | 3.1 | 2.9 | 2.3 | 0.9 | 0.8 | 0.2 | 3.8 | 3.7 | 2.7 | 2.9 | 0.6 | 1.2 |
| | FK | - | 1.3 | 1.5 | 2.4 | - | 1.5 | 2.7 | 1.4 | - | 1.2 | 1.2 | 0.3 |
| | **PG-DLM** | - | 2.2 | 3.1 | 0.2 | - | 1.8 | 1.5 | 2.3 | - | 1.0 | 1.1 | 0.2 |

## D.4 CHG: ABLATIONS ON HYPER-PARAMETERS FOR TABLE 2

Table 9: Controlled text generation accuracies across reward functions (CoLA, Toxicity, Sentiment) on the MDLM base model, comparing PG-DLM against the baseline method FK Steering (FK) under varying compute budgets (columns) and configuration settings (rows). **Columns** labeled 4 – 64 correspond to NFEs normalized by the total number of denoising steps $T$, i.e. NFE/$T$. **Rows** labled (*,*,*) indicates, respectively: partial reward sampling methods (Beam, Random), diffusion backward processes (MDLM, ReMDM), and resample frequency (20, 5). Fixed parameters: generated length $L = 50$, total denoising timesteps $T = 1024$, $\beta = 0.1$, number of partial reward samplers $\phi = 1$. For PG-DLM, we use $m = 1$. Thus the compute budget is controlled by the number of samples $k$ for both FK Steering and PG-DLM.

| Method | CoLA ↑ | | | Toxicity ↑ | | | Sentiment ↑ | | |
|---|---|---|---|---|---|---|---|---|---|
| | 4 | 16 | 64 | 4 | 16 | 64 | 4 | 16 | 64 |
| **FK Steering (FK)** | | | | | | | | | |
| (Rand, MDLM, 20) | $48.1 \pm 1.6$ | $79.0 \pm 4.3$ | $87.1 \pm 1.9$ | $3.8 \pm 1.0$ | $39.8 \pm 3.7$ | $86.1 \pm 1.1$ | $37.4 \pm 1.2$ | $91.3\pm3.4$ | $99.7 \pm 0.3$ |
| (Rand, MDLM, 5) | $48.4 \pm 3.2$ | $76.2 \pm 0.4$ | $83.1 \pm 4.8$ | $3.4 \pm 0.2$ | $34.0 \pm 3.4$ | $76.8 \pm 1.1$ | $33.6 \pm 3.7$ | $89.2 \pm 1.5$ | $98.9 \pm 0.5$ |
| (Rand, ReMDM, 5) | $87.4 \pm 1.7$ | $93.6 \pm 1.0$ | $92.9 \pm 1.3$ | $16.9 \pm 0.7$ | $89.7 \pm 1.3$ | $97.6 \pm 0.2$ | $67.7 \pm 2.8$ | $97.9 \pm 0.7$ | $99.4 \pm 0.2$ |
| (Beam, MDLM, 5) | $66.6 \pm 1.7$ | $94.8 \pm 0.2$ | $97.8 \pm 1.0$ | $11.2 \pm 1.1$ | $81.9 \pm 3.0$ | $96.8 \pm 1.0$ | $57.6 \pm 5.9$ | $94.2 \pm 0.8$ | $99.2 \pm 0.2$ |
| (Beam, ReMDM, 5) | $91.7 \pm 0.9$ | $97.8 \pm 0.7$ | $97.5 \pm 0.2$ | $24.6 \pm 0.7$ | $95.4 \pm 0.7$ | $98.7 \pm 0.3$ | $72.3 \pm 4.3$ | $96.1 \pm 1.1$ | $99.2 \pm 0.2$ |
| **PG-DLM** | | | | | | | | | |
| (Random, MDLM, 5) | $29.8 \pm 3.1$ | $80.0 \pm 1.2$ | $89.4 \pm 1.1$ | $1.3 \pm 0.0$ | $26.8 \pm 2.7$ | $75.1 \pm 2.7$ | $12.8 \pm 2.0$ | $82.7 \pm 2.1$ | $99.1 \pm 0.5$ |
| (Random, ReMDM, 5) | $74.8 \pm 3.0$ | $97.4 \pm 0.7$ | $98.7 \pm 0.7$ | $1.6 \pm 0.5$ | $84.8 \pm 0.8$ | $96.4 \pm 1.8$ | $24.7 \pm 1.2$ | $96.0 \pm 0.9$ | $99.6 \pm 0.5$ |
| (Beam, MDLM, 5) | $37.3 \pm 2.4$ | $88.0 \pm 1.0$ | $96.8 \pm 0.5$ | $1.3 \pm 0.5$ | $78.8 \pm 2.0$ | $97.2 \pm 1.2$ | $21.8 \pm 1.7$ | $94.4 \pm 0.5$ | $99.0 \pm 0.3$ |
| (Beam, ReMDM, 5) | $77.3 \pm 2.0$ | $97.3\pm 0.9$ | $99.1 \pm 0.5$ | $1.4 \pm 0.7$ | $91.1\pm1.0$ | $98.1 \pm 1.1$ | $23.8 \pm 2.2$ | $96.2 \pm 1.3$ | $99.1\pm 0.2$ |

## D.5 CHG: QUALITATIVE EXAMPLES

| Method | Generated Output |
|--------|------------------|
| best-of-$n$ | |
| | • Once upon a time, this was one of my favorite taglines in Indie Match Match :The impossible we overcome Those that we escape The Impossible were our face. The Impossible were our face |
| | • The chicken is still really amazing after consuming the amount is parox Imagine had orange soup. The soup has very low sugar release. The whole concept of this is that it helps as an antioxidant. It's an antioxidan |
| | • The lake went up through the fields, the hills cracked, and fell to the sea. Heaven came clean, the wind sang like the mountains: BRAND BLOOD Now black, skin on cold, Ice white |
| FK | |
| | • Once upon a time,was one of the coolest and most beautiful colors of all time. Nowadays, this color is among my favorite colors of all time. Let me show you guys with some pictures of what my favorite colors look like |
| | • The chicken was extremely tender and flavorful. There was a nice crunchiness to chicken wings on top. I do prefer to eat chicken wings when they are a little smaller and less crunchy. I also enjoyed keeping the wings in the refrigerator |
| | • The lake temperature is colder in the spring, which allows you to use the water easier. At a depth above the current lake level, you can find the most beautiful thermal lakes in North America. The lakes are brilliant |
| PG-DLM | |
| | • Once upon a time, the openmindedness and diversity of the universe was one of the pillars of our success, and continues to be. Today, we welcome the diversity and nature of the universe, and embrace it as a |
| | • The chicken burger really live up to the deli's spot for the dish. The fried chicken wings really make it an addition of the menu due to their cute goo and I LOVE THEM! The burger isn't the best |
| | • The lake itself is totally potable and there are plenty of holes in the middle of the lake. It is perfect for any kind of tradition of mountaineering adventure.The lake is also used as a point of contact and |

Table 10: Qualitative comparison of generated sequences under a positive sentiment reward

## E USE OF LLMS

We only use Large Language Models for polish writing.

