# OpenReview forum: "Inference-Time Scaling of Diffusion Language Models with Particle Gibbs Sampling"
_ICLR.cc/2026/Conference — Submitted to ICLR 2026_

### Official Review · Reviewer_xrTZ · 2025-10-28

**Soundness:** 3
**Presentation:** 3
**Contribution:** 3
**Rating:** 6
**Confidence:** 2

**Summary:**

The authors propose a technique for inference-time scaling of diffusion language models using particle Gibbs sampling. Their approach is empowered with conditional Sequential Monte-Carlo (SMC) technique which allows to do step refinement of sampling trajectory. On top of that, they propose to do additional sequential trajectory-level refinements after SMC steps which allows to scale up amount of steps at inference and bring additional improvements to the reward-guided sampling. The paper have extensive ablations over the hyperparameters which control the amount of inference-time compute as well as extensive empirical evidence of the advantage of their approach.

**Strengths:**

1. The paper is clearly written

2. The proposed method is simple and easy-to-understand

3. The paper contains extensive ablations of hyperparameters which control the amount of compute at inference, making it very easy for a reader to understand which of these parameter matter in practice.

4. The paper contains extensive experimental evidence demonstrating that this approach works very well in practice, compared to the baselines

**Weaknesses:**

* While the paper provides convergence and variance bounds in Section 3.4, there are no proofs for both Theorem 1 and Theorem 2. Can the authors provide full proofs for these both theorems?

* Theorem 2 implies that the variance is divided by both k and m, but in Table 4, we see that ESS is mainly high for high values of k. In fact for m=1, k=64, it has the highest values, whereas for k=8,m=8, it has the lowest values. Theorem 2 implies that these should behave in the same way, since variance is divided by (k * m = 64 in both cases. Why is there such a discrepancy? Is the derivation in theorem 2 correct?

**Questions:**

1. Can you provide proofs for both, Theorem 1 and Theorem 2?

2. Why is there a discrepancy in Table 4? (see Weaknesses). Is the derivation for the variance in theorem 2 correct?

---

> ### Author Response · Authors · 2025-11-22
>
> We thank the reviewer for their positive feedback! We are encouraged that they found the paper clearly written, the proposed method simple, and the experiments extensive in demonstrating the effectiveness of our approach. Below, we address the reviewer’s comment in detail.
>
> # 1. Proofs of Theorem 1 and Theorem 2.
>
> Full proofs for both theorems were included in the original submission and the revised version. Specifically:
> - Theorem 1 is proved in **Appendix B.1**.
> - Theorem 2 is proved in **Appendix B.2**.
>
> To make this clearer, we've added a pointer to the relevant appendices in line 248 of the revised version.
>
> # 2. Clarification on ESS Interpretation in Table 4.
>
> We appreciate the reviewer’s careful reading and insightful observation. We would like to clarify that the **Effective Sample Size (ESS)** is computed per iteration ($\text{Iter} = 1, \cdots, m$), based on the normalized weights of the $k$ samples at that iteration: $$\text{ESS}=1/\sum_{i=1}^k w_i^2.$$
> This implies that ESS **is upper-bounded by the number of samples $k$** for each setting, regardless of the number of iterations $m$. For example, in the setting $m=1,k=64$, ESS is upper-bounded by 64; in contrast, for $m=8,k=8$, ESS is upper-bounded by 8.
>
> To provide a more direct comparison across different settings, we additionally compute the total $m \times \text{ESS}$ at the final iteration. As shown below, the values are comparable across settings and increase slightly with larger $m$, which aligns with the theoretical prediction:
> | Setting |  $m \times \text{ESS}$ |
> |--|--|
> |$m=1,k=64$| 60.2 $\pm$ 5.3 |
> |$m=2,k=32$|61.1$\pm$ 6.3|
> |$m=4,k=16$|62.1 $\pm$ 4.9 |
> |$m=8,k=8$|62.2 $\pm$ 4.7|
> We have added a clarification in the caption of Table 4 to make it more explicit in the revised version.
>
> We believe the derivation in Theorem 2 is correct. The variance bound follows the standard analysis of particle-Gibbs and Monte Carlo estimators with resampling. We hope the clarification above helps address the reviewer’s concern.

---

> > ### Comment · Reviewer_xrTZ · 2025-11-24
> > **Response**
> >
> > I would like to thank the authors for their response and for clarifying my confusion.
> >
> > I am keeping my current score.

---

### Official Review · Reviewer_7ri2 · 2025-10-28

**Soundness:** 4
**Presentation:** 4
**Contribution:** 2
**Rating:** 6
**Confidence:** 3

**Summary:**

The paper introduces a new method for inference-time scaling of diffusion language models. The proposed method is the application of particle Gibbs to this setting; it repeats the SMC sampling process iteratively, where at each iteration it uses the best trajectory from the previous iteration as a reference, so as to calibrate the SMC weights across iterations. These higher level iterations provide an axis for scaling in addition the existing axes of number of particles, time-steps and Monte Carlo runs for partial reward estimation inside SMC. Empirical evaluation suggests that this new axis can be beneficial at higher NFEs when compared to other axes.

**Strengths:**

-- The paper is clearly written, there’s adequate background material and it is easy to follow.
-- Theoretical results on consistency and variance are interesting.
-- The scope of experiments is reasonable and the results, although not very strong, do support the general conclusions of the paper.
-- The paper’s empirical study of different scaling axes is an interesting contribution.

**Weaknesses:**

-- The main weakness seems to be about the cost-benefit of the proposed idea: the additional cost is sequential in nature and when compared to parallel scaling (e.g., number of particles), there seems to be a marginal benefit at large scale; it is hard to see even this in figure 2, which is supposed to show the benefit of larger m at larger NFE (am I right?)

-- Partial reward estimate uses beam sampling in scaling applied to reward estimation of SMC. A better alternative is to use more accurate reward estimates through (partial) unrolling the reverse diffusion for reward estimation, for example in \phi steps. Although this is a sequential process, and therefore not as efficient, so is the proposed scaling in the number of iterations. Can you consider this kind of scaling for reward estimation in your study?

-- Prior work search based methods [1,2,3,4] such as MCTS applied to diffusion also revisit trajectories for refinement, and they have been applied to both continuous and discrete diffusion and language. While the paper does generally a reasonable job of discussing related literature, I think it should also discuss these works and adjust its claim about its novelty in improving trajectories in multiple passes.

Minor:

-- Table captions say NFE is up to 128 while it is up to 64
-- Fix reference in algorithm 1 is repeated

[1] Zhang, Xiangcheng, et al. "Inference-time scaling of diffusion models through classical search." arXiv preprint arXiv:2505.23614 (2025).

[2] Jain, Vineet, et al. "Diffusion Tree Sampling: Scalable inference-time alignment of diffusion models." arXiv preprint arXiv:2506.20701 (2025).

[3] Tang, Sophia, et al. "TR2-D2: Tree Search Guided Trajectory-Aware Fine-Tuning for Discrete Diffusion." arXiv preprint arXiv:2509.25171 (2025).

[4] Guo, Yingqing, et al. "Training-free guidance beyond differentiability: Scalable path steering with tree search in diffusion and flow models." arXiv preprint arXiv:2502.11420 (2025).

**Questions:**

-- Please let me know if there is a misunderstanding under “weaknesses”.

-- What does “discretization error” mean in the context of theorem 2?

-- Can you justify the following? “reward model has a similar computational cost to the denoiser”

---

> ### Author Response · Authors · 2025-11-22
>
> We thank the reviewer for their thoughtful feedback. We are encouraged that they found our paper clearly written, theoretical results interesting, and experiments supportive, and they appreciated our scaling axes analysis. Below, we address specific points and have incorporated all feedback in the revised version (updated PDF).
>
> > **W1. "there seems to be a marginal benefit at large scale; it is hard to see even this in figure 2"**
>
> Thank you for raising this point. We agree that the gain appears marginal in Figure 2 at large NFE. The original submission included Tables 1 and 6 to better illustrate the benefits.
>
> To clarify, we have added new experiments in Section 5.4. As you noted, PG‑DLM has a sequential nature, thus it enables adaptive compute allocation (e.g., do more iterations only if the reward on $x_0​$ is low), which outperforms parallel methods like SMC, especially on variable-difficulty tasks like math reasoning. On GSM8K with LLaDA, PG-DLM (adapt) achieves **92.80%** accuracy at 8 average samples, vs. SMC's **90.45%**. This highlights our method's advantage in adaptive allocation, which is unavailable in SMC.
>
> We show some experiment results here; full plots are in Section 5.4 (revised version).
>
>
> Comparing the accuracy of different methods for LLaDA on GSM8k under a fixed budget
>
> | num-samples     |     1 |     2 |     4 |     8 |    16 |    32 |
> |-----------------|------:|------:|------:|------:|------:|------:|
> | greedy-decoding | 78.14 |       |       |       |       |       |
> | best-of-n       | 62.68 | 75.63 | 83.69 | 88.15 | 90.83 |       |
> | SMC             |      | 81.96 | 88.55 | 90.45 | 92.32 | 93.40 |
> |PG-DLM              |       |  | | **91.36** | **93.25** | **94.01** |
>
> Accuracy results of PG-DLM (adapt) and the average number of samples
> | averaged num-samples |  1.94 |  2.98 |  4.72 |  7.60 | 13.88 | 24.82 |
> |----------------------|------:|------:|------:|------:|------:|------:|
> | PG-DLM (adapt) | 84.76 | 90.07 | 92.12 | 92.80 | 94.24 | 94.47 |
>
> > **W2. “Can you consider this kind of scaling for reward estimation in your study?”**
>
> Thank you for the thoughtful suggestion. We agree that more accurate multi-step unrolling could complement our approach, and we have added a discussion of this alternative in Line 250 and Line 359 in the revised version of the paper. Multi-step unrolling ($\tau > 1$) for reward estimation is orthogonal to our partial estimation via $\phi$ candidate $x_0$ samples. While NFEs may match when $\phi = \tau$, sampling $\phi$ candidates batches efficiently, unlike sequential unrolling. We use one-step greedy sampling ($\phi=1, \tau=1$) for its efficiency..
>
> > **W3. "Prior work search based methods [1,2,3,4] … adjust its claim about its novelty in improving trajectories in multiple passes."**
>
> Thank you for highlighting these works. We have added discussion of [1, 2, 4] in Section 6. Note that [3] was posted after ICLR deadline.
>
> Notably, [1, 2] incorporate search mechanisms that can revisit clean data $x_0$​ via backtracking in the search tree, while [4] performs tree search without explicit refinement of $x_0$​.
>
> In contrast, our approach introduces a particle-based resampling mechanism that iteratively refines $x_0$ under a probabilistic inference framework, a capability that was not studied in particle-based methods before. Both particle-based resampling and search-based methods offer distinct inference-time scaling alternatives.
>
> We have expanded the introduction (lines 52–62 in the revised version) to acknowledge these methods that perform trajectory-level refinement:
>
> *"More recent search-based methods (Zhang et al., 2025a; Jain et al., 2025) achieve trajectory-level refinement by revisiting full generations via backtracking in a search tree."*
>
> We clarify our contribution immediately afterward:
>
> *"In contrast, we introduce the first particle-based framework that performs trajectory-level refinement through iterative resampling of complete trajectories within an SMC algorithm, which enables probabilistic inference and adaptive compute allocation."*
>
> > **W4.1"Table captions say NFE is up to 128 while it is up to 64"**
>
> Thank you for catching this. We have corrected the caption of Table 2 in the revised version.
>
> > **W4.2"Fix reference in algorithm 1 is repeated"**
>
> Thank you for the comment. We assume this refers to the repeated lines 7 and 13 (lines 6 and 12 in the updated PDF). This repetition is intentional: line 7 uses $\bar{x}$ to denote the examples before resampling, while line 13 uses ${x}$ to denote the resampled examples that proceed to the next timestep. Please let us know if we have misunderstood the concern.

---

> ### Author Response · Authors · 2025-11-22
>
> > **Q1. "Please let me know if there is a misunderstanding under “weaknesses”."**
>
> We believe two points may reflect misunderstandings: W3 (some work is posted after the deadline) and W4.2 (the repeated line is intentional). We find the other points to be valuable, and we have addressed them with additional experiments and revisions.
>
> > **Q2. "What does “discretization error” mean in the context of theorem 2?"**
>
> The discretization error refers to approximating continuous-time CTMC ($t \in [0,1]$) [5] with finite steps $T$; as $T \to \infty$, the discretization error vanishes. For discrete-time targets like MDLM [6], no such assumption is needed, and we have updated the notation accordingly (line 247, footnote 1).
>
> [5] A continuous time framework for discrete denoising models. Campbell et al. 2022.
>
> > **Q3. "Can you justify the following? “reward model has a similar computational cost to the denoiser”"**
>
> In our setup, the ~700M-parameter MDLM denoiser matches the ~500M-parameter reward models (Toxicity/Sentiment/CoLA), so one denoising step is approximately one reward evaluation. This justifies our (1 + $\phi$) scaling factor for partial estimation. For larger denoisers (e.g., 7B) with small rewards, the reward cost is negligible. We have updated Table 1 comparisons.
>
> **Closing Remarks**
>
> We hope that our responses address your concerns. If any point remains unclear or requires further discussion, we would be very happy to provide additional explanations or experiments.

---

> > ### Comment · Reviewer_7ri2 · 2025-11-25
> > **comparison to partial unrolling**
> >
> > I thank the authors for their clarifications and updates to the paper that reflect the reviews. While I will closely follow the discussion with other reviewers, I still would like clarification on the comparison with partial unrolling.
> >
> > The rebuttal argues that such a comparison is unnecessary because the two choices are "orthogonal". However, if I have a compute budget to spend on _sequential_ scaling, I could choose between partial unrolling versus the particle Gibbs. Unless there is an obvious point that I'm missing, I believe it makes sense to provide such a comparison. This is particularly useful given the underwhelming performance improvements reported in some settings.

---

> ### Author Response · Authors · 2025-11-25
>
> Thank you for the response and engaged discussion.
>
> > "The rebuttal argues that such a comparison is unnecessary because the two choices are "orthogonal"..."
>
> We would like to clarify that we did *not* state (nor imply) that such a comparison is unnecessary.
> We described the two directions as orthogonal, as they reduce different sources of error ( $\tau$ for discretization error vs. $\phi$ for Monte-Carlo error) and can be combined. To eliminate any ambiguity, we have revised the paper to clarify this point explicitly and now note that deeper partial unrolling is a complementary and promising scaling strategy.
>
> > "If I have a compute budget to spend on sequential scaling, I could choose between partial unrolling versus the particle Gibbs"
>
> Absolutely agreed. With a fixed sequential compute budget, one can allocate FLOPs across diffusion time steps $T$, number of particles $m$, and unrolling depth $\tau$. In this work, we deliberately follow prior diffusion literature by fixing one-step unrolling ($\tau = 1$) and study the trade-off between $T$, the number of Gibbs passes $k$, and particles $m$. We have added text acknowledging that increasing $\tau$ is a promising direction under the same budget.
>
> Specifically in the paper, we said:
>
> *"In practice, we set $\tau=1$ for efficiency following prior works. However, studying the scaling behavior of $\tau$ is an interesting and promising complementary future direction."*
>
> > "This is particularly useful given the underwhelming performance improvements reported in some settings."
>
> We would greatly appreciate it if the reviewer could indicate which specific tasks or settings appeared underwhelming.
>
> We view the GSM8K experiments as a key example of strong performance gains (+2%~3% accuracy over the best baseline), and note that for tasks like CoLA and Toxicity, PG-DLM outperforms the baselines under the same compute budget.
>
> If there are particular cases where the improvements seem unclear or limited, we would be happy to address them more directly.
>
> Thank you again for the careful reading and valuable suggestions; they have noticeably improved the clarity of the paper.

---

### Official Review · Reviewer_9miR · 2025-11-04

**Soundness:** 1
**Presentation:** 3
**Contribution:** 2
**Rating:** 2
**Confidence:** 4

**Summary:**

The paper introduces PG-DLM, a particle Gibbs sampling method for discrete diffusion LMs. The method is grounded in sequential refinement using conditional SMC, by using the highest reward trajectory as “reference” for the next round to explore variations around this high-reward sample. This allows the practitioner to scale sequential refinement iterations in addition to particle size. This method enjoys the theoretical guarantees of existing SMC methods. Experiments using MDLM and LLaDA backbones show that scaling iterations and particle counts improve final scores on reward like toxicity classifier, linguistic acceptability classifier, and sentiment classifier. PG-DLM performs on par with existing methods.

**Strengths:**

- Sequential refinement is an important scaling axis, and this method is a practical method to allow scaling beyond particle count.
- The method and algorithm are presented clearly and are easy to understand.
- PG-DLM enjoys the theoretical convergence guarantees of SMC, so it is a principled method.

**Weaknesses:**

My main concerns revolve around fair comparison with baselines, evaluation metrics, as well as clarifications on certain statements and conclusions drawn from the experiments.

- **Unfair comparison with baselines.** Section 5.1 says that all methods including PG-DLM resample every 20 steps, but Table 7 says PG-DLM resamples every 5 steps (as opposed to 20 for the baselines). Please update the main text to correctly reflect these differences. Perhaps even more crucially, PG-DLM uses the ReMDM backward process and compares it against disadvantaged baselines using vanilla backward process (this is again somewhat hidden and only mentioned in the appendix). For a fair comparison, please provide results by keeping factors other than the PG-DLM algorithm consistent to isolate gains. Looking at Figure 6, it seems like removing ReMDM from PG-DLM results in a substantial drop in performance.
- **Evaluation shows high rewards, but not overall quality and coherency.** Since the goal is to sample from the reward-tilted posterior (Section 2.2), simply reporting high reward is not enough - it needs to be balanced with high likelihood under the model. In other words, gains in toxicity/CoLA/sentiment score or large PPL drops can be accompanied with a drop in overall coherency, quality and faithfulness to the prompt, since these reward functions are notoriously bad evaluators [1,2] and can be hacked by simply repeating certain words. I suggest the authors (1) report evaluation using humans or LLM-as-judge, since it has been shown that they align better with human judgements [3], and (2) provide qualitative examples to show the method results in still coherent text that respond to the prompt appropriately.
- **Statistical significance of results.** All results are reported on a very small set of 15 prompts, which is not sufficient to draw general conclusions. Also, it seems all results are reported for a single seed. I *strongly* suggest the authors report results + variance for multiple seeds (and, where relevant, across prompts) to improve the reliability of the results.
- **Theorem 1 and 2 are known results.** The authors note that since PG-DLM is essentially conditional SMC, existing guarantees apply to their method as well. I find it a bit odd to state these guarantees as theorems without citing what works they were adapted from. I suggest rewriting them as remarks, providing references to the existing results and explaining how they are still valid when applied to the diffusion setting.
- **Theory–practice gap and strong assumptions.** Asymptotic guarantees require unbiased posterior means and no discretization error. These assumptions are asserted (e.g., for MDLM/LLaDA) but not empirically checked, while most experiments use finite T and a biased reward estimate (which they call beam sampling). This weakens the bridge between theory and the recommended operating point
- **Overstating novelty.** The claim of being the “first trajectory-level inference-time sampling method for discrete diffusion models with formal convergence and variance guarantees” is overstated, since prior works [4,5] also propose trajectory-level methods with formal guarantees; these relevant methods should also be discussed and contrasted with PG-DM.
- **Conclusions drawn from experiments.**  Some of the conclusions and discussion in the text needs more clarifications given the provided plots.
    - Section 4 discusses that increasing the number of Gibbs iterations effectively improve accuracy at mid-to-high NFE regions, but I don’t see it from the plots in Figure 2 since across $m=1,2,4,8$, the performance at higher NFEs seems to be more or less the same, and at lower NFEs increasing $m$ is worse (as noted by authors).
    - I am having trouble seeing that increasing $k$ helps more than increasing $T$ in Section 4. If we look at Figure 4 and Appendix C, it seems increasing beyond $k=8,16$ there is very little gain. However, in certain ranges there is large gain in performance by increasing denoising steps. I think these plots are not very well-suited for showing trade-offs between two different parameters, I suggest using scatter plots with $k,T$ on the two axes.

*[1] Holtzman, Ari, et al. "The Curious Case of Neural Text Degeneration." International Conference on Learning Representations, 2020.*

*[2] Hashimoto, Tatsunori B., Hugh Zhang, and Percy Liang. "Unifying Human and Statistical Evaluation for Natural Language Generation." Proceedings of the 2019 Conference of the North American Chapter of the Association for Computational Linguistics: Human Language Technologies, Volume 1 (Long and Short Papers). 2019.*

*[3] Liu, Yang, et al. "G-Eval: NLG Evaluation using Gpt-4 with Better Human Alignment." Proceedings of the 2023 Conference on Empirical Methods in Natural Language Processing. 2023.*

*[4] Jain, Vineet, et al. "Diffusion Tree Sampling: Scalable inference-time alignment of diffusion models." The Thirty-ninth Annual Conference on Neural Information Processing Systems, 2025.*

*[5] Guo, Yingqing, et al. "Training-free guidance beyond differentiability: Scalable path steering with tree search in diffusion and flow models." arXiv preprint arXiv:2502.11420 (2025).*

**Questions:**

- The introduction motivates PG-DLM by saying it performs “trajectory-level refinement”. But why is this desirable, and what is the underlying problem being solved here? Later, they say the “one-shot” generation of SMC is a limitation, without elaborating why.
- The authors propose a beam sampling method to estimate intermediate rewards, however, it performs the same as random sampling for $\phi > 1$. And for $\phi=1$, as noted by the authors, it is equivalent to greedy/argmax decoding. Can the authors explain the benefit of this beam sampling method?
- Could the authors explain what they meant in Line 51: SMC and particle-based methods  “operate within a single denoising trajectory” when they do have multiple trajectories that get resampled at regular intervals?
- Line 157 “While the reward-weighted conditional structure can be derived formally from an RL perspective, similar formulations have also been adopted as sampling heuristic”. Can the authors explain how these formulation have been developed without explicit connection to RL objectives? All of these results and the authors’ own derivation in Appendix B use soft-Bellman equations and other RL concepts.
- Line 205 “Samples evolve as independent trajectories interacting only via resampling, limiting inter-sample correlations.” SMC *does* induce dependence via resampling and shared weights, could the authors explain what is meant here?
- I am having trouble understanding the values of NFEs reported in the experiments. Table 2 and 3 report results for 1,…,64 NFEs, while a single denoising process takes 1024 NFEs (for MDLM) and 50 NFEs (for LLaDA). So are these per-step-NFEs? Will the authors consider using total NFEs everywhere for consistency and simplicity?

**Details Of Ethics Concerns:**

The paper proposes a general-purpose inference-time alignment method for discrete diffusion models. One set of experiments focuses on increasing toxic content in natural language generation. While this is not the main focus of the paper, I believe this should be accompanied by an ethics statement.

---

> ### Author Response · Authors · 2025-11-22
>
> We thank the reviewer for their detailed and thorough feedback. We are encouraged that they found our method important, principled, and practical, and appreciated the clarity of the presentation.
> In response to the reviewer’s concerns regarding fair comparison with baselines, evaluation metrics, and clarification of experimental conclusions, we have made several updates in the revised version. Major changes are highlighted in **red** for clarity:
>
> 1. We have revised the theorem statement for improved clarity and precision. (**Section 3.4**)
> 2. We have updated the experimental section with more detailed descriptions and additional ablations. (**Section 5.1, Appendix D.4**)
> 3. We have added new experiments on math reasoning tasks to better demonstrate the effectiveness of our approach. (**Section 5.4**)
>
> We clarify and address each concern in detail below.

---

> ### Author Response · Authors · 2025-11-22
>
> # W1. Comparisons with Baselines
> > "Please update the main text to correctly reflect these differences."
>
> Thank you for pointing this out. While the difference in backward process and resampling frequency was included in the appendix, we have now made them explicit in the main text for clarity (Line 402).
>
> > "PG-DLM uses the ReMDM backward process and compares it against disadvantaged baselines using vanilla backward process. For a fair comparison, please provide results by keeping factors other than the PG-DLM algorithm consistent to isolate gains."
>
> We clarify that in the original submission, *we followed the standard practice of reproducing prior work in the literature by following their original configurations.* For PG-DLM, PG-DLM, we incorporated improved techniques (e.g., ReMDM backward process) alongside our novel sampling. The same techniques can also be used to improve SMC baselines. As the reviewer noted, Figure 6 already ablated backward processes.
>
> To further address this concern, we *have included new ablations in Appendix D.4* comparing PG-DLM and SMC across backward processes (ReMDM and MDLM), sampling methods (Random, Beam), resample frequency, and number of iterations. We observe that our method still has a benefit over the baseline. Ablations on PG-DLM are provided here, full results are in Appendix D.4:
>
> |                 | CoLA (4)           | CoLA (16)        | CoLA (64)        | Toxicity (4)    | Toxicity (16)    | Toxicity (64)    | Sentiment (4)    | Sentiment (16)   | Sentiment (64)   |
> |-----------------|--------------------|------------------|------------------|-----------------|------------------|------------------|------------------|------------------|------------------|
> | (Random, MDLM)  | 29.8 $\pm$   3.1   | 80.0 $\pm$   1.2 | 89.4 $\pm$   1.1 | 1.3 $\pm$   0.0 | 26.8 $\pm$   2.7 | 75.1 $\pm$   2.7 | 12.8 $\pm$   2.0 | 82.7 $\pm$   2.1 | 99.1 $\pm$   0.5 |
> | (Random, ReMDM) | 74.8 $\pm$   3.0   | 97.4 $\pm$   0.7 | 98.7 $\pm$   0.7 | 1.6 $\pm$   0.5 | 84.8 $\pm$   0.8 | 96.4 $\pm$   1.8 | 24.7 $\pm$   1.2 | 96.0 $\pm$   0.9 | 99.6 $\pm$   0.5 |
> | (Beam, MDLM)    | 37.3 $\pm$   2.4   | 88.0 $\pm$   1.0 | 96.8 $\pm$   0.5 | 1.3 $\pm$   0.5 | 78.8 $\pm$   2.0 | 97.2 $\pm$   1.2 | 21.8 $\pm$   1.7 | 94.4 $\pm$   0.5 | 99.0 $\pm$   0.3 |
> | (Beam, ReMDM)   | 77.3$\pm$   2.0 | 97.3$\pm$ 0.9  | 99.1 $\pm$ 0.5 | 1.4 $\pm$ 0.7 | 91.1$\pm$1.0   | 98.1 $\pm$ 1.1 | 23.8 $\pm$ 2.2 | 96.2 $\pm$ 1.3 | 99.1$\pm$ 0.2  |
>
> # W2. Evaluation
> > "imply reporting high reward is not enough - it needs to be balanced with high likelihood under the model."
>
> Thank you for the suggestion. In the original submission, Figure 3 shows the trade-off between accuracy and perplexity for PG-DLM, which shows that PPL does not drop as increasing iterations $m$.
>
> > "reward functions are notoriously bad evaluators [1,2] and can be hacked by simply repeating certain words."
>
> We agree that these reward functions, such as Toxicity and Sentiment, are easy to hack. However, this is orthogonal to our study, as this limitation is shared across all inference methods. And our goal is to show that PG-DLM  follows the reward signal more effectively than other inference methods.
>
> To further address this concern, we *include additional experiments on GSM8K*, where rewards correspond to task correctness rather than keywords, which are verifiable rewards and can not be hacked. In these settings, we show that PG-DLM achieves stronger accuracy under fixed compute budgets and supports adaptive compute allocation. The (adapt) variant achieves the best accuracy under all compute budget with a significant margin. Detailed results and setups are in Section 5.4.
>
> > "it has been shown that they align better with human judgements"
>
> For GSM8K experiments, we use task accuracy as the evaluation metric, which directly reflects correctness and aligns with human judgment.
>
> > "provide qualitative examples to show the method results in still coherent text that respond to the prompt appropriately."
>
> We have added qualitative examples in Appendix D.5 to illustrate that PG-DLM produces coherent and prompt-relevant generations.
>
>
> # W3. Statistical Significance
> > "All results are reported on a very small set of 15 prompts"
>
> We clarify that the controllable generation experiments use 15 prompts, but each with 20 generations, following prior work.
>
> In newly added experiments (Section 5.4), we report results on GSM8K (1319 test prompts), addressing concerns about generalization to larger prompt sets.
>
> > "it seems all results are reported for a single seed."
>
> We clarify that in the original submission, the main Table 2 reported mean performance over 3 random seeds, with standard deviations provided in Appendix D.3. While a pointer to this was included in the main text, we have now made it more explicit for clarity (line 406 in the updated version).  Both Table 9 and Figure 7 (in revised version) show that the results are separated.

---

> ### Author Response · Authors · 2025-11-22
>
> # W4. Theorems
>
> We have revised Theorems 1 and 2 and isolated a Lemma in the revised version for better clarity.
>
> > "I find it a bit odd to state these guarantees as theorems without citing what works they were adapted from."
>
> Thank you for the comment. *In the original submission, we cited original works in line 245, 254, and 255 (Lines 245, 258, 259 in the revised version).* These references were intended to acknowledge that the general consistency and variance bounds for SMC-based methods are known results.
>
> > "Theorem 1 and 2 are known results."
>
> We agree that the general consistency for SMC is well-known. We built on these results; however, *our goal is to clarify how  this applies for discrete diffusion models, and this exact theorem does not exist and is not immediately obvious*, thus it is important to adapt it to this particular setting. As the reviewer noted in W5, it is important to show how these assumptions hold in theory and how they relate to practical design choices.
>
> For Theorem 2, while the variance bound builds on known SMC literature (which we cited), *we further provide a variance analysis specific to the reward-weighted target distribution for diffusion models.* In particular, we relate the estimator variance to the variance of the reward function under the proposal and target distributions. If this connection has been made previously, we would greatly appreciate a reference and will cite it accordingly.
>
> # W5. Theory-practice Gap
>
> Thank you for pointing this out. We have updated Theorems 1 and 2 to *clarify this explicitly via Lemma 1* (Line 250).
>
> > "these assumptions are asserted (e.g., for MDLM/LLaDA) but not empirically checked" "explaining how they are still valid when applied to the diffusion setting."
>
> We now discuss the theory–practice gap in more detail (Lines 299), highlighting how the assumptions relate to practical design choices.
>
> Specifically:
>
> "Lemma 1 holds for discrete diffusion models such as MDLM and LLaDA. However, in practice, we approximate partial rewards using a small number of $\phi$ samples, each generated with only one denoising step. While this deviates from the asymptotic setting, the convergence and variance bounds still provide valuable insight into how PG-DLM performance scales with different factors, such as $m,k,T,\phi$, which we study empirically in Section 5."
>
> # W6. Novelty
>
> Thank you for pointing out these relevant works. They are both search-based methods, and our method is resampling-based, performing trajectory-level refinement within a probabilistic inference framework.
>
> We acknowledge that [4] proposes a trajectory-level method by backtracking in the search tree to revisit clean data $x_0$​ via, while [5] performs tree search without explicit refinement of $x_0$​.
>
> In contrast, our approach introduces a *particle-based resampling mechanism* that iteratively refines $x_0$. Unlike search, our resampling-based approach offers an alternative that scales more naturally to settings where search becomes intractable, such as large action spaces and search trees.
>
> We have revised Sections 1 and 6 to clarify this distinction and more accurately state our contribution. See line 74 and line 522 in the revised version.

---

> ### Author Response · Authors · 2025-11-22
>
> # W7. Experiments
>
> > "the performance at higher NFEs seems to be more or less the same"
>
> Thank you for raising this point. We agree that the gain appears marginal in Figure 2 at large NFE. The original submission included Tables 1 and 6 to better illustrate the benefits.
>
> To clarify, *we have added new experiments in Section 5.4.* PG‑DLM has a sequential nature, thus it enables adaptive compute allocation (e.g., do more iterations only if the reward on $x_0​$ is low), which outperforms parallel methods like SMC on difficult tasks like math reasoning. On GSM8K with LLaDA, PG-DLM (adapt) achieves **92.80%** accuracy at 8 average samples, vs. SMC's **90.45%**. This highlights our method's advantage in adaptive allocation, which unavailable in purely parallel schemes like SMC. Some numbers are illustrated here; full plots are in Section 5.4.
>
>
> Comparing the accuracy of different methods for LLaDA on GSM8k under a fixed budget
>
> | num-samples     |     1 |     2 |     4 |     8 |    16 |    32 |
> |-----------------|------:|------:|------:|------:|------:|------:|
> | greedy-decoding | 78.14 |       |       |       |       |       |
> | best-of-n       | 62.68 | 75.63 | 83.69 | 88.15 | 90.83 |       |
> | SMC             |      | 81.96 | 88.55 | 90.45 | 92.32 | 93.40 |
> |PG-DLM              |       |  | | **91.36** | **93.25** | **94.01** |
>
> Accuracy results of PG-DLM (adapt) and the average number of samples
> | averaged num-samples |  1.94 |  2.98 |  4.72 |  7.60 | 13.88 | 24.82 |
> |----------------------|------:|------:|------:|------:|------:|------:|
> | PG-DLM (adapt) | 84.76 | 90.07 | 92.12 | 92.80 | 94.24 | 94.47 |
>
>
> > "If we look at Figure 4 and Appendix C, it seems increasing beyond k=8,16  there is very little gain."
>
> In Figure 4 and Appendix C, the x-axis represents NFEs (by changing $T$), different curves show different $k$, and the y-axis shows accuracy. The curve for $k=32$ consistently outperforms others, except in the settings where accuracy is already saturated (e.g., Figure 4.C). In such cases, increasing $k$ provides a marginal gain simply because performance is already near 100%. However, in these same settings, increasing T leads to even smaller gains.
>
>
> > "However, in certain ranges there is large gain in performance by increasing denoising steps."
>
> We appreciate the pointer and would welcome clarification on which specific plots this refers to. In the non-saturated regime (lower NFEs), we observe that increasing $k$ (i.e., moving from lighter to darker curves) consistently improves accuracy more than increasing $T$ (moving right on the x-axis). In the saturated regime, performance plateaus for both axes, but increasing $k$ typically remains slightly better.

---

> ### Author Response · Authors · 2025-11-22
>
> # Q1
>
>
> > “The introduction motivates PG-DLM by saying it performs “trajectory-level refinement”. But why is this desirable, and what is the underlying problem being solved here?”
>
> Thank you for the question. Trajectory-level refinement leverages full generations $x_0$ and introduces a new scaling axis: number of iterations, as shown in our original submission.
>
> More importantly, our added GSM8K experiments (Section 5.4) demonstrate that it enables adaptive compute allocation, versus SMC's fixed upfront budget. This achieves the best accuracy under all compute budgets with a significant margin
>
> > "they say the “one-shot” generation of SMC is a limitation, without elaborating why."
>
> However, for SMC the compute budget (number of samples $k$) must be determined in advance. Once generation is complete, the trajectories cannot be revisited or refined. Therefore, it lacks adaptive compute allocation abilities.
>
> # Q2
>
> > "Can the authors explain the benefit of this beam sampling method?"
>
> Certainly. While beam sampling and random sampling perform similarly for larger $\phi$, beam (greedy decoding) performs significantly better than random sampling when $\phi=1$, as shown in Figure 5.
>
> We use $\phi=1$ because increasing $\phi$ improves partial reward estimation but raises NFEs (scaled by $(1+\phi)$). Under fixed budgets, boosting $k$ outperforms boosting $\phi$ as shown in Figure 5. Thus, $\phi=1$ with greedy decoding offers the best trade-off.
>
> # Q3
> > "SMC and particle-based methods “operate within a single denoising trajectory” when they do have multiple trajectories that get resampled at regular intervals?"
>
> We use *trajectory* to refer to the full denoising path from $x_T$ to $x_0$. SMC maintains multiple samples, each representing a full trajectory, and resamples noisy latents $x_t$ at intermediate.
>
> While SMC involves multiple trajectories, they are parallel and not sequentially refined on the same $x_0$.
> To clarify this, we have revised the sentence to (line 50):
>
>  “However, these methods (e.g., SMC) maintain multiple parallel samples, each following a single denoising trajectory $x_T,\cdots,x_0$, sampled step-by-step from $t=T$ to $t=0$, with resampling at intermediate timesteps.”
>
> # Q4
>
> > "Can the authors explain how these formulation have been developed without explicit connection to RL objectives?"
>
> Certainly. Prior works [6, 7] define the target distribution directly as the reward-tilted distribution $p(x_0|c)\cdot \exp(r(c,x_0))$, without linking to RLHF objection (cf. Equation 4 in our paper).
>
> Besides, they use rewards differences across timesteps $r(c, x_{t-1}) - r(c, x_t)$ as a heuristic to guide generations (e.g., Section 3.3 in [6]), without showing a connection to the underlying target distribution.
>
> In contrast, as the reviewer noted, we provided a formal derivation from an RL perspective that explains why using partial reward differences is effective with reward-weighted inference, offering a principled foundation for this heuristic (cf. Equation 6 in our paper).
>
> [6] A General Framework for Inference-time Scaling and  Steering of Diffusion Models. Singhal et al. 2025.
>
> [7] Human preference score v2: A solid benchmark for evaluating human preferences of text-to-image synthesis. Wu et al. 2023.
>
> # Q5
>
> > "“Samples evolve as independent trajectories interacting only via resampling, limiting inter-sample correlations.” SMC does induce dependence via resampling and shared weights, could the authors explain what is meant here?"
>
> Thank you for pointing this out. We agree that SMC induces dependencies among particles through resampling and shared importance weights. This is why our original sentence stated that samples *"interact only via resampling"*, acknowledging that this is the source of dependence.
>
> For clarity, we revised "independent" to "parallel":
> “Samples evolve as parallel trajectories interacting only via reweighting and resampling, limiting inter-sample correlations between them.”
>
> # Q6
>
>
> > "Table 2 and 3 report results for 1,…,64 NFEs …So are these per-step-NFEs?"
>
> Yes, the values in Tables 2 and 3 are per-step NFEs. To improve clarity, we have updated Tables 2 and 3 to use NFE/T explicitly:
>
> "Columns labeled 1–64 correspond to NFEs normalized by the total number of denoising steps $T$, i.e. $\text{NFE}/T$."
>
> Since MDLM and LLaDA use different numbers of denoising steps (1024 and 50, respectively), but other configurations are the same, we use per-step NFEs for clarity.
>
> # Ethics Concern
>
> Thank you for raising this important point. The toxicity red-teaming experiment was intended purely as a diagnostic to verify reward following, following prior works [6]. We have updated the ethics statement to clarify this intent and explicitly discourage misuse. Our method is reward-agnostic and can be used to promote beneficial behaviors when paired with appropriate reward models.

---

> ### Author Response · Authors · 2025-11-22
>
> # Closing Remarks
>
> We hope that our responses sufficiently address your concerns. Thank you again for your thoughtful feedback, and we welcome any further suggestions you may have.

---

> > ### Comment · Reviewer_9miR · 2025-11-23
> >
> > Thank you for your detailed response. The new results, clarifications on various statements, and the updated ethics statement are appreciated.
> >
> > ### Comparison with baselines
> > The comparison with the baseline using the same denoising process is appreciated. To respond to the author's clarification, the standard practice when comparing different methods is to keep factors other than the novel algorithm consistent, so that the merits of the proposed approach can be judged fairly without any other confounding factors.
> >
> > After examining the results in Section D.4, it seems that when performing an apples-to-apples comparison, *FK-steering consistently outperforms PG-DLM by a large margin* across all three reward functions (for both random denoising and ReMDM denoising). Could the authors point specifically to where PG-DLM has benefits over the baseline?
> >
> > Minor note - since my main concern was specifically about comparing with baselines, it might have been prudent to post baseline comparison results in the rebuttal response for the convenience of other reviewers and the AC.
> >
> > ### Evaluation
> > Thank you for adding results on GSM8K and the qualitative examples, which show PG-DLM still generates coherent responses. However, I must stress that contrary to the author's statement, simply following the reward signal is not the goal of any inference-time algorithm. Instead, the goal is to sample from a reward-tilted posterior by balancing both reward and model likelihood. Some existing methods, despite using easy-to-hack reward functions, can manage to avoid over-optimization by careful design of the algorithm [DTS; Jain et al. 2025, DAS; Kim et al. 2025].
> >
> > ### Theorems and theory-practice gap
> > Thank you for adding the additional analysis. My comment about adding relevant citations was to specify which prior results were specifically used in Theorems 1 and 2 (by stating them as Theorem 1 [adapted from XXX et al.], for example). However, after the revisions in the paper, this is a relatively minor point.
> >
> > ### Novelty and discussion on existing work
> > Thank you for the added discussion in Section 6. While I appreciate PG-DLM is distinct from prior search-based methods since it performs sequential refinement within a particle-based framework, there are two points I would like to make in this regard:
> > - The introduction (line 52 onwards) is still missing a discussion on these relevant search-based methods, while asserting that PG-DLM enables trajectory-based refinement and introduces a new scaling axis. As acknowledged by the authors, these tree-search methods do perform trajectory-based refinement, and [DTS; Jain et al. 2025] specifically discusses scaling along two axes, and also constitutes a “sampling” method (line 74). Therefore, in light of these recent works, the statements in the introduction still overstate novelty, and it should include a discussion of these works to appropriately position this work wrt existing works.
> > - I would like to respectfully push back against the author's statement that particle-based methods scale better than search methods. The prior works described above have (1) shown to be effective in high-dimensional spaces with across a large range of compute budgets, and (2) in very large action-spaces, particle-based methods do not enjoy any specific advantages over search-based methods since they are also limited by the parallel particle count.
> >
> > ### GSM8K experiments
> > Do these experiments use PG-DLM with ReMDM and SMC with a random denoising process? If so, please report the results, keeping the denoising process consistent in line with the above discussion.
> >
> > ### Increasing k vs increasing denoising steps
> > I was specifically referring to:
> > - In Figure 4 (c), for certain budget ranges, the curve of $k=32$ is slightly worse than $k=8,16$.
> > - Figure 4 (b), where increasing denoising steps for $k=4$ results in performance going up from 0.2 to 0.8.
> > - Similarly, in Appendix C, Figures 8 and 9 show that certain cases for $k=4$ and $k=8$, increasing denoising steps results in a large increase in score.
> >
> > My point was not to say that increasing $k$ does not help improve performance, or that scaling $T$ is strictly better than scaling $k$, I was simply stating that, based on some of these plots, it might not be as clear-cut as stated in the paper.
> >
> > ### Connection to RL objectives
> > Thank you for the clarification. Some existing works have indeed stated the reward-tilted distribution and the corresponding SMC weights without connecting them to theory; my point was that this connection has been known in literature for a while [Uehara et al. 2024], and the previous works simply failed to respect this connection to RL literature. I appreciate the authors doing their due diligence and the derivation in Appendix B.1 is okay for the sake of completeness, but it largely follows a similar derivation in [Uehara et al. 2024, Jain et al. 2025], and this derivation should not be stated as a new insight.

---

> ### Author Response · Authors · 2025-11-25
>
> Thank you for the careful and engaged discussion. We appreciate your constructive review. Your deep and precise questions helped us address several subtle but important points. We address each concern in the following.
>
> # GSM8K experiments
> > “Do these experiments use PG-DLM with ReMDM and SMC with a random denoising process? “
>
> **No. The GSM8K experiments for both PG-DLM and SMC use the same denoising process**: LLaDA’s native random denoising process.
>
>
> # Comparison with baselines
> We sincerely thank the reviewer for the careful examination of Appendix D.4. We first answer the question “where PG-DLM has benefits over the baseline” and then explain the results in Appendix D.4.
>
> ## The benefit of PG-DLM
> > “Could the authors point specifically to where PG-DLM has benefits over the baseline?”
>
> 1. Adaptive compute allocation
>
> PG-DLM naturally supports adaptive compute allocation. **On GSM8K, the adaptive variant consistently outperforms the strongest SMC baseline by over 2% accuracy at matched compute** (an apples-to-apples comparison). (Figure 7 and Table W7 in the first response).
>
> - ~2 effective samples: PG-DLM's **84.76%**  vs SMC's **81.96%**
> - ~8 effective samples: PG-DLM's **92.80%** vs SMC's **90.45%**
>
> 2. Multi-iteration refinement ($m>1$)
>
> Beyond adaptive allocation, PG-DLM can allocate compute from more samples $k$ to more refinement iterations $m>1$. For example, in GSM8K:
>
> -   32 effective samples: PG-DLM ($k=8,m=4$)'s **94.01%** vs SMC ($k=32$)'s **93.40%**
> -   16 effective samples: PG-DLM ($k=8,m=2$)'s **93.25%** vs SMC ($k=16$)'s **92.32%**

---

> ### Author Response · Authors · 2025-11-25
>
> # Comparison with baselines (continued)
> ## Explanation of Appendix D.4
>
> > "After examining the results in Section D.4, it seems that when performing an apples-to-apples comparison, FK-steering consistently outperforms PG-DLM by a large margin across all three reward functions."
>
> Here is the table comparing PG-DLM ($m=1$) and FK-steering with different partial-reward estimation (Beam/Random), different backward processes (MDLM/ReMDM), under different normalized NFEs (4, 16, 64). (Note that here "Random" refers to the random sampling partial reward estimator, and "MDLM" refers to the random backward process.)
> |                 | CoLA (4)           | CoLA (16)        | CoLA (64)        | Toxicity (4)    | Toxicity (16)    | Toxicity (64)    | Sentiment (4)    | Sentiment (16)   | Sentiment (64)   |
> |-----------------|--------------------|------------------|------------------|-----------------|------------------|------------------|------------------|------------------|------------------|
> | FK-Steering|
> | (Random, MDLM)    | 48.4 $\pm$   3.2 | 76.2 $\pm$   0.4     | 83.1 $\pm$   4.8     | 3.4 $\pm$   0.2  | **34.0 $\pm$   3.4** | _76.8 $\pm$   1.1_ | 33.6 $\pm$   3.7 | **89.2 $\pm$   1.5** | _98.9 $\pm$   0.5_ |
> | (Random, ReMDM)   | 87.4 $\pm$   1.7 | 93.6 $\pm$   1.0     | 92.9 $\pm$   1.3     | 16.9 $\pm$   0.7 | **89.7 $\pm$   1.3** | _97.6 $\pm$   0.2_ | 67.7 $\pm$   2.8 | **97.9 $\pm$   0.7** | _99.4 $\pm$   0.2_ |
> | (Beam, MDLM)    | 66.6 $\pm$   1.7 | **94.8 $\pm$   0.2** | _97.8 $\pm$   1.0_   | 11.2 $\pm$   1.1 | _81.9 $\pm$   3.0_   | _96.8 $\pm$   1.0_ | 57.6 $\pm$   5.9 | _94.2 $\pm$   0.8_   | _99.2 $\pm$   0.2_ |
> | (Beam, ReMDM)   | 91.7 $\pm$   0.9 | _97.8 $\pm$   0.7_   | 97.5 $\pm$   0.2     | 24.6 $\pm$ 0.7   | **95.4 $\pm$   0.7** | _98.7 $\pm$   0.3_ | 72.3 $\pm$   4.3 | _96.1 $\pm$   1.1_   | _99.2 $\pm$   0.2_ |
> | PG-DLM ($m=1$)|
> | (Random, MDLM)  | 29.8 $\pm$   3.1 | **80.0 $\pm$   1.2** | **89.4 $\pm$   1.1** | 1.3 $\pm$   0.0  | 26.8 $\pm$   2.7     | _75.1 $\pm$   2.7_ | 12.8 $\pm$   2.0 | 82.7 $\pm$   2.1     | _99.1 $\pm$   0.5_ |
> | (Random, ReMDM) | 74.8 $\pm$   3.0 | **97.4 $\pm$   0.7** | **98.7 $\pm$   0.7** | 1.6 $\pm$   0.5  | 84.8 $\pm$   0.8     | _96.4 $\pm$   1.8_ | 24.7 $\pm$   1.2 | 96.0 $\pm$   0.9     | _99.6 $\pm$   0.5_ |
> | (Beam, MDLM)    | 37.3 $\pm$   2.4 | 88.0 $\pm$   1.0     | _96.8 $\pm$   0.5_   | 1.3 $\pm$   0.5  | _78.8 $\pm$   2.0_   | _97.2 $\pm$   1.2_ | 21.8 $\pm$   1.7 | _94.4 $\pm$   0.5_   | _99.0 $\pm$   0.3_ |
> | (Beam, ReMDM)   | 77.3 $\pm$   2.0 | _97.3$\pm$ 0.9_      | **99.1 $\pm$ 0.5**   | 1.4 $\pm$ 0.7    | 91.1$\pm$1.0         | _98.1 $\pm$ 1.1_   | 23.8 $\pm$ 2.2   | _96.2 $\pm$ 1.3_     | _99.1$\pm$ 0.2_    |
>
> 1. The table reports results with $m=1$. By design, for both MDLM and ReMDM backward, **when "Beam" is disabled and $m=1$, PG-DLM reduces to a single iteration of standard conditional SMC with a fixed reference trajectory. This is intentionally almost identical to FK-steering/SMC (differing only by the reservation of one reference particle, yielding k−1 "fee" particles instead of k).**
>
> 2. The performance using the same partial-reward estimation (Beam/Random), different backward processes (MDLM/ReMDM) (see the above table) **confirms implementation correctness**.
>     1. The performance gap at the lowest budget (normalized NFE=4, thus $k=2$) is therefore expected due to the one-particle overhead: PG-DLM has one "free" particle and FK has two.
>     2. The performance gap disappears at larger budgets (normalized NFE > 16): in the 24 apples-to-apples comparisons, PG-DLM and FK-steering are close in 13 cases (overlapping mean $\pm$ std, italicized) and FK better in 6, PG-DLM better in 5 (bolded).
>
> 3. The consistent advantage of **PG-DLM (Beam, $m=1$) over FK-steering (Random) demonstrates the value** of the proposed partial reward estimator as our method. The beam-based partial-reward estimator **(Beam) itself is a novel method (Section 4.2)** that is simple but effective and benefits any SMC steering method, including FK-steering.
>
> We highlight that **_PG-DLM’s distinctive advantages appear only when its unique scaling axes are exercised: when the additional scaling axis $m>1$ or adaptive compute is leveraged._** Thus, **we refer to the GSM8K experiments as the major results to demonstrate the benefits.** In settings like CoLA, where accuracy saturates near 100%, $m=1$ is sufficient.

---

> ### Author Response · Authors · 2025-11-25
>
> # Evaluation
> > “However, I must stress that contrary to the author's statement, simply following the reward signal is not the goal of any inference-time algorithm”
>
> Thank you for the comment. We clarify that we did not state that the goal is simply to follow the reward signal. We fully agree that the objective is to sample from the reward-tilted posterior, which balances both the reward and the model likelihood. Our formulation in Section 3 and Algorithm 1 is designed to reflect this balance.
>
> # Theorems and theory-practice gap
>
> Thank you for the suggestion. We have now revised the text to make the connection with prior works more explicit by labeling the results as “Theorem 1 [adapted from Andrieuetal.,2010]” and “Theorem 2 [adapted from Andrieuetal.,2010; Chatterjee&Diaconis,2018]” as recommended.
>
> # Novelty and discussion on existing work
> We sincerely thank the reviewer for the careful reading and constructive feedback. We address two points in the following:
>
> 1. Discussion of search-based methods
>
> We have now expanded the introduction (lines 52–62 in the revised version) to explicitly acknowledge recent search-based methods that perform trajectory-level refinement:
>
> *"More recent search-based methods (Zhang et al., 2025a; Jain et al., 2025) achieve trajectory-level refinement by revisiting full generations via backtracking in a search tree."*
>
> We clarify our contribution immediately afterward:
>
> *"In contrast, we introduce the first particle-based framework that performs trajectory-level refinement through iterative resampling of complete trajectories within an SMC algorithm, which enables probabilistic inference and adaptive compute allocation."*
>
> 2. Claim about scaling superiority
>
> We fully agree. We appreciate the reviewer’s note and fully agree with the reviewer. *We never claimed (nor intended to claim) that particle-based methods scale better than search-based methods.* Both paradigms are highly effective and have different strengths. Our focus is exclusively to contribute to a deeper understanding of the particle-based scaling, not to argue superiority.
>
> We hope these clarifications and revisions fully address the reviewer’s concerns.
>
> # Connection to RL objectives
>
> > “my point was that this connection has been known in literature for a while [Uehara et al. 2024], and the previous works simply failed to respect this connection to RL literature.”
>
> We agree that the connection between reward-tilted distributions and RL objectives has been known, and we cited prior works accordingly. And we agree with the reviewer that prior works have failed to respect this connection, for example, (Singhal et al. 2025) describes SMC weights as “heuristics” without citing RL-based interpretations. Our motivation was to make this connection explicit and to emphasize its importance.
>
>
> > “it largely follows a similar derivation in [Uehara et al. 2024, Jain et al. 2025], and this derivation should not be stated as a new insight.”
>
> We agree, and we clarify that we did not state this derivation as a new insight, and we explicitly cited Uehara et al. 2024. In the paper (Section 3.1), we stated *“Building on prior works in the continuous setting(Ueharaetal., 2024a;b), we derive the tractable formulation for these conditionals in the discrete setting”*
>
> Our goal was to adapt this existing derivation to the discrete diffusion setting for completeness and clarity.

---

> ### Author Response · Authors · 2025-11-25
>
> # Increasing k vs increasing denoising steps
>
> Thank you for the clarification and for pointing out specific examples.
> - In Figure 4(c), we agree that in certain budget ranges, the curve for $k=32$ is slightly worse than for $k=8,16$; besides, performance saturates in that range (e.g., accuracy is already near 100%). Our main claim is not that increasing $k$ always helps, but that given the same total NFE budget, increasing $k$ is *generally* more effective in most cases.
>
>
> - In Figure 4(b), increasing $T$ for $k=4$ leads to a gain from 0.2 to 0.8, but for the same NFE budget, curves for $k=8, 16, 32$ already exceed 0.8, achieving accuracy ~0.95 (**bolded** in the table below). This suggests that allocating compute toward $k$ rather than spending all in $T$ yields greater benefits in that regime. We've included a small table (below) to help illustrate the numerical comparison.
>
> The table used to plot Figure 4(b) is organised such that each column corresponds to a fixed NFE, and each row shares the same $k$ with varying $T$  and corresponding accuracy.
> |        | $T*k=2^9$ | $T*k=2^{10}$ | $T*k=2^{11}$ | $T*k=2^{12}$ | $T*k=2^{13}$ | $T*k=2^{14}$ |          |          |          |      |
> |--------|-----------|--------------|--------------|--------------|--------------|--------------|----------|----------|----------|------|
> | $k=4$  | $T=128$   | $T=256$      | $T=512$      | $T=1024$     | $T=2048$     | $T=4096$     |          |          |          |      |
> |        |      0.17 |         0.29 |         0.47 |         0.64 |         0.73 |         **0.80** |          |          |          |      |
> | $k=8$  |           | $T=128$      | $T=256$      | $T=512$      | $T=1024$     | $T=2048$     | $T=4096$ |          |          |      |
> |        |           |         0.61 |         0.74 |         0.86 |         0.90 |         **0.96** |     0.97 |          |          |      |
> | $k=16$ |           |              | $T=128$      | $T=256$      | $T=512$      | $T=1024$     | $T=2048$ | $T=4096$ |          |      |
> |        |           |              |         0.81 |         0.91 |         0.94 |         **0.95** |     0.96 |     0.98 |          |      |
> | $k=32$ |           |              |              | $T=128$      | $T=256$      | $T=512$      | $T=1024$ | $T=2048$ | $T=4096$ |      |
> |        |           |              |              |         0.90 |         0.95 |         **0.98** |     0.99 |     0.96 |     0.98 | 0.98 |
>
> - Similarly, in Appendix C (Figures 8 and 9), increasing denoising steps indeed improves performance, but again, using the same NFE budget to also increase $k$ often leads to larger overall gains.
>
>
> > "it might not be as clear-cut as stated in the paper."
>
> Thank you for the clarification. The paper stated that:
>
> *"...increasing $k$ generally provides greater benefits in both settings..."*
>
> which was not intended as a clear-cut claim, but rather to express a trend observed in most cases.
>
> That said, we appreciate the reviewer’s observation. We've updated the text to explicitly acknowledge this and better reflect the trade-off:
>
> *"...increasing $k$ generally provides greater benefits, though in some cases, e.g., when the performance saturates as in Figure 4 (c), smaller $k$ can be better"*
>
> # Closing Remarks
> We hope these revisions and clarifications fully address your concerns. Should any point remain unclear, we would of course be happy to provide further details or experiments.

---

### Meta-Review · Area_Chair_ZLWv · 2026-01-05

**Summary:**

The paper applies particle Gibbs to sample from the reward tilted measure of a diffusion language model. This work is natural given the revived interest in sequential monte carlo for sampling in diffusion models. There were questions about the related work, comparisons, evaluations (a focus on rewards, rather than the defined sampling problem), importance of the underlying theorems, and clarity on when the approach proposed warrants use. Several of these things were clarified in the author response, such as the proposed method's merit being adaptive compute at the cost of being sequential instead of parallel. Overall given the current literature, revisiting particle gibbs for this problem is a good idea, but the presentation of the paper and experimental results need to be sharpened.

**Reviewer Concerns:**

9miR's concerns: related work, evaluations, and clarity on when the method should be used

7ri2's concerns: question about the sequential nature of the computation versus the parallel version

xrTZ's concerns: questions about theorems and their implications

**Reviewer Scores:**

xrTZ: stays the same

7ri2: stays the same

9miR: might increase to a 4

---

### Decision · Program_Chairs · 2026-01-26

Reject